# Detection of genetic variation and base modifications at base-pair resolution on both DNA and RNA

Zhen Wang[1], Jérôme Maluenda[1], Laurène Giraut[1], Thibault Vieille[1], Andréas Lefevre [1], David Salthouse[1], Gaël Radou[1], Rémi Moulinas[1], Sandra Astete[1], Pol D'Avezac[1], Geoff Smith[1], Charles André [1], Jean-François Allemand [2,3], David Bensimon[2,3,4], Vincent Croquette[2,3,5], Jimmy Ouellet[1,6] & Gordon Hamilton [1,6 ✉]

Accurate decoding of nucleic acid variation is critical to understand the complexity and regulation of genome function. Here we use a single-molecule magnetic tweezer (MT) platform to identify sequence variation and map a range of important epigenetic base modifications with high sensitivity, specificity, and precision in the same single molecules of DNA or RNA. We have also developed a highly specific amplification-free CRISPR-Cas enrichment strategy to isolate genomic regions from native DNA. We demonstrate enrichment of DNA from both *E. coli* and the *FMR1* 5'UTR coming from cells derived from a Fragile X carrier. From these kilobase-length enriched molecules we could characterize the differential levels of adenine and cytosine base modifications on *E. coli*, and the repeat expansion length and methylation status of *FMR1*. Together these results demonstrate that our platform can detect a variety of genetic, epigenetic, and base modification changes concomitantly within the same single molecules.

[1] Depixus SAS, 3/5 Impasse Reille, 75014 Paris, France. [2] Laboratoire de physique de L'École normale supérieure de Paris, CNRS, ENS, Université PSL, Sorbonne Université, Université de Paris, Paris 75005, France. [3] IBENS, Département de biologie, École normale supérieure, CNRS, INSERM, PSL Research University, 75005 Paris, France. [4] Department of Chemistry and Biochemistry, UCLA, 607 Charles E Young Drive East, Los Angeles 90095, USA. [5] ESPCI Paris, PSL University, 10 rue Vauquelin, 75005 Paris, France. [6] These authors jointly supervised this work: Jimmy Ouellet, Gordon Hamilton. ✉email: gh@depixus.com

Next-generation sequencing (NGS) has enabled a revolution in our understanding of genomics. Current NGS instrument systems are highly scalable, flexible and generate accurate sequence data that is valuable in many different applications[1–3]. Progress has been rapid due to the dramatic reduction in sequencing costs and continuous improvements to data quality. Despite these advances, determining the entire genetic sequence of a sample with short-read NGS systems has proven too expensive for many routine research and translational experiments. In addition, epigenetic and long-range structural data are also typically missed, with many NGS assays at best only providing indirect measurements of genome function. For transcript analysis, for example, the conversion of RNA to cDNA erases the numerous modifications on RNA bases and creates quantification bias after multiple cycles of amplification[4].

To meet these challenges, different NGS workflows have been developed, each with its own benefits and trade-offs. Some of the most widely used workflows to reduce sequencing costs rely on capturing specific regions of the genome, using either PCR amplification or affinity purification, and then sequencing these focused libraries using short read technology[5–8]. Compared to full genome sequencing, these protocols dramatically reduce per-sample costs as multiple samples can be pooled and multiplexed; however, because these methods involve sample amplification, this benefit comes at the penalty of loss of epigenetic information and the introduction of bias into the results[9]. Recently amplification-free Cas9-based enrichment strategies have been attempted by several groups[10–12] but typically only achieve an enrichment of 20-fold to 60-fold (compared to >10,000-fold for PCR)[13].

By contrast, the ability to generate long-range genomic information has required the advent of longer read-length, single-molecule sequencing approaches, for example using nanopores (Minion, Oxford Nanopore)[13] and zero-mode waveguides (Sequel, Pacific Biosciences)[14,15]. Although these platforms have proven to be useful for closing gaps in short read sequencing, and in providing important long-range structural information, they lack the per-read accuracy of NGS, and to date have only characterized a very limited range of epigenetic modifications. To complement these existing systems, we are developing a universal platform for genomic and epigenomic analysis that records the complexity of information on native nucleic acid molecules of both DNA and RNA. The platform is based on magnetic tweezers (MT) technology, which has been widely used for elucidating the function of DNA polymerases and accessory proteins required for DNA replication[16–20].

In a typical experiment on our platform, DNA or RNA molecules are converted into hairpin structures. Each hairpin is then attached by one of its free ends to a micron-scale paramagnetic bead and anchored by the other end to a planar glass surface. When a magnetic force (of greater than approximately 15 pN) is applied to the tethered beads, these hairpins mechanically open (unzip) to become single-stranded. They then reform (re-zip) again with relaxation of the force (Fig. 1a)[21]. When ligands that bind to nucleic acids are introduced into the system, their bound presence on the molecule can disrupt hairpin unzipping or rezipping, and the position of these transient blockages can be precisely mapped to the sequence of the hairpins (Fig. 1b). As the process is non-destructive, the same hairpin molecules can be opened and closed many times in a single experiment. This makes it possible to determine both the on-rates of different binding ligands (based on the probability of observing the bound state), and their off-rates (from the average time of the transient hairpin blockage). The accuracy of locating binding positions and determining binding kinetics increases with increasing numbers of open-close cycles. Moreover, changing the ligand enables the successive detection of different nucleic acid sequences and/or different base modifications on the same molecule. To our knowledge, this feature is unique among single molecule genomic technologies and allows both the mapping of multiple ligands on the same molecule (either simultaneously or sequentially), and the application of error correction techniques to improve analytical accuracy.

Here we demonstrate the performance of a new genetic analysis platform based on MTs that allows the accurate analysis of long molecules of DNA and RNA. Previous versions of MT instruments relied on diffraction from a single light source to measure the Z-position of paramagnetic beads, with single base accuracy reported only for fragments of DNA less than 80 base pairs[22]. To allow the analysis of longer molecules with greater accuracy, a new instrument was designed with two important changes[23]. Firstly, a system for detecting and tracking bead position was developed based on stereo darkfield interferometry (SDI) in which paramagnetic beads are illuminated by a pair of light sources to generate a correlated set of diffraction fringes. Accurate measurement of the relative displacement of these fringes in the X axis corresponds to the position of the bead in the Z axis[23] (Patent EP3181703B1). In addition, a system was developed to provide a high degree of temperature stability to allow experiments of longer duration which are required for the analysis of multiple features on these molecules. We used this new instrument to detect the underlying sequence structure and a range of base modifications together in model DNA and RNA templates. In addition, we developed a novel enrichment method to target genomic regions in native DNA samples without the need for amplification. This allowed us to analyze adenine and cytosine methylation of specific E. coli genomic regions together in the same individual molecules, and to characterize both the underlying structural variation (trinucleotide repeats) and epigenetic modification of single molecules of native DNA molecules from the clinically important gene, FMR1. Further improvements to the accuracy, usability and throughput of this MT platform will expand the application of these methodologies across many areas of genomic research to reveal the details underlying the complexity of genetic control and regulation.

## Results

### Characterization of a new generation of MT instrument with single base resolution.

To expand the application of MTs to genomic studies, it was important to assess the new SDI instrument system for its positional precision with longer nucleic acid molecules. For these studies, we constructed a 600 bp double-stranded DNA hairpin (Supplementary Fig. 1) with four binding sites for an 11-mer oligonucleotide and tracked bead positions across 100 cycles of zipping and rezipping under magnetic force (Fig. 1b). The positions of all four blocking positions were within 1 base pair of the expected locations (Fig. 1c). We then extended the analysis to a longer 5 kb hairpin (Supplementary Fig. 2) and found the resolution of the instrument to be 1 bp or less for molecular lengths up to 1.5 kb, with precision decreasing as a function of molecular length. We attribute the improved positional accuracy of the SDI instrument to lower instrument noise (0.9 bp vs. 2 bp) (Supplementary Fig. 3). Moreover, better temperature control allowed us to maintain the temperature of the sample within 1 millikelvin over 60 h, as well as reduce thermal drift by magnet thermalization (Supplementary Fig. 4). Overall, this new version of the MT instrument offers improved precision and higher throughput than previous instruments.

### Identification and mapping of DNA base modifications.

Having shown the potential of the new MT instrument to accurately

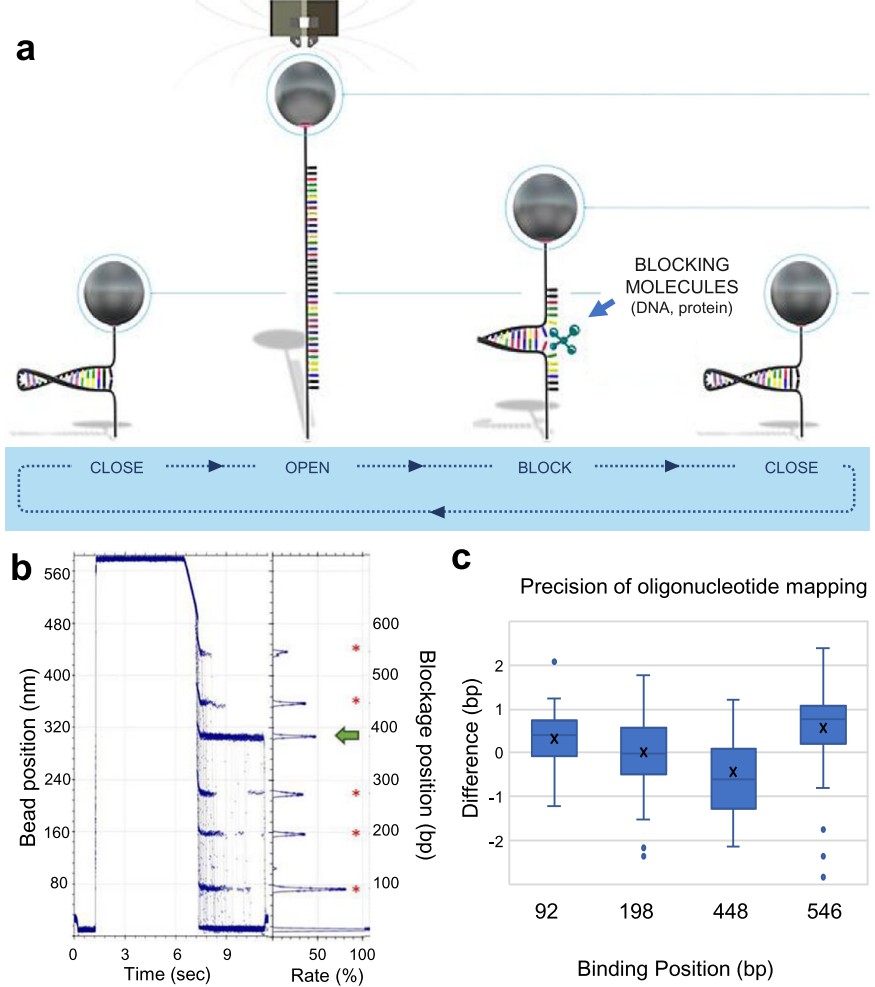

**Fig. 1 Analytical approach and precision of the new MT platform. a** Schematic representation of a typical MT cycle. When the force increases, by approaching the magnets to the sample, the DNA hairpin molecule is denatured and binding molecules (either oligonucleotides or proteins) bind to the ssDNA nucleic acid. Upon reduction in force, the hairpin reforms and transient blockages occur at the binding positions. **b** All the cycles are overlaid, and a cumulative histogram of the blocking positions is built. In this example, a 11-base reference oligonucleotide that binds five times (red asterisks) was injected at the same time as an antibody against m5C modification (green arrow). **c** Mapping of reference blocking positions of a 11 DNA base oligonucleotide on a 600 bp hairpin ($n = 80$ individual molecules). The average experimental positions versus the expected positions were between ±one base for the majority of the molecules. Whisker boxes represent 50% of the points with the average as a line within the box and the median as a cross.

map oligonucleotide blocking positions, we assessed whether antibodies selected for binding to different DNA base modifications could also block hairpin reformation and whether these blockages could identify and locate the position of these modifications. We assembled a DNA hairpin from chemically synthesized oligonucleotides that contained seven different base modifications at defined locations: (5-methylcytosine (m5C), 5-hydroxymethylcytosine (hm5C), 5-carboxylcytosine (ca5C), 5-formylcytosine (f5C), 3-methylcytosine (m3C), N6-methyladenine (m6A), and 8-oxoguanine (8-oxoG)) (Fig. 2a and Supplementary Fig. 2). We then tested commercially available antibodies for their sensitivity and specificity of detection of these modified bases (Fig. 2a).

For 6 out of the 7 modifications, we identified an antibody that could detect the expected modification with a sensitivity greater than 95% of the molecules analyzed (the number of detected modifications over the total expected, Table 1). Furthermore, by correlating these blockages to the positions of a series of reference oligonucleotides, all the antibodies tested mapped to their respective base modification within an average of 1 bp (Fig. 2d). For four of the antibodies tested (those raised against hm5C, ca5C,

m6A, and 8-oxoG), the binding was highly specific, and the blockages corresponded to the expected position of the cognate antigen (Fig. 2b). For two other antibodies (raised against m5C and f5C), we detected the expected base, but also identified a second binding position mapping to the position of one of the other base modifications (anti-m5C cross-reacted with hm5C, and anti-f5C cross-reacted with ca5C, Fig. 2b).

We tested whether we could distinguish true positive blockage events from those occurring through cross-reactivity by comparing the binding times and frequencies (proportion of cycles for which a blockage occurred) for both the m5C and f5C antibodies. For anti-m5C, we found that by applying a simple threshold for binding time and frequency we could cluster the true positives from off-target interactions (Fig. 2c). This suggested that the interaction of the anti-m5C antibody to m5C was of a higher affinity than the binding to hm5C. By contrast, no such simple cut-off could be applied to the anti-f5C antibody. However, because we can test different antibodies sequentially on the same DNA molecules, we could cross-correlate blockage data to identify the correct base modification. For f5C, for example, comparing the blockages obtained with anti-f5C and anti-ca5C

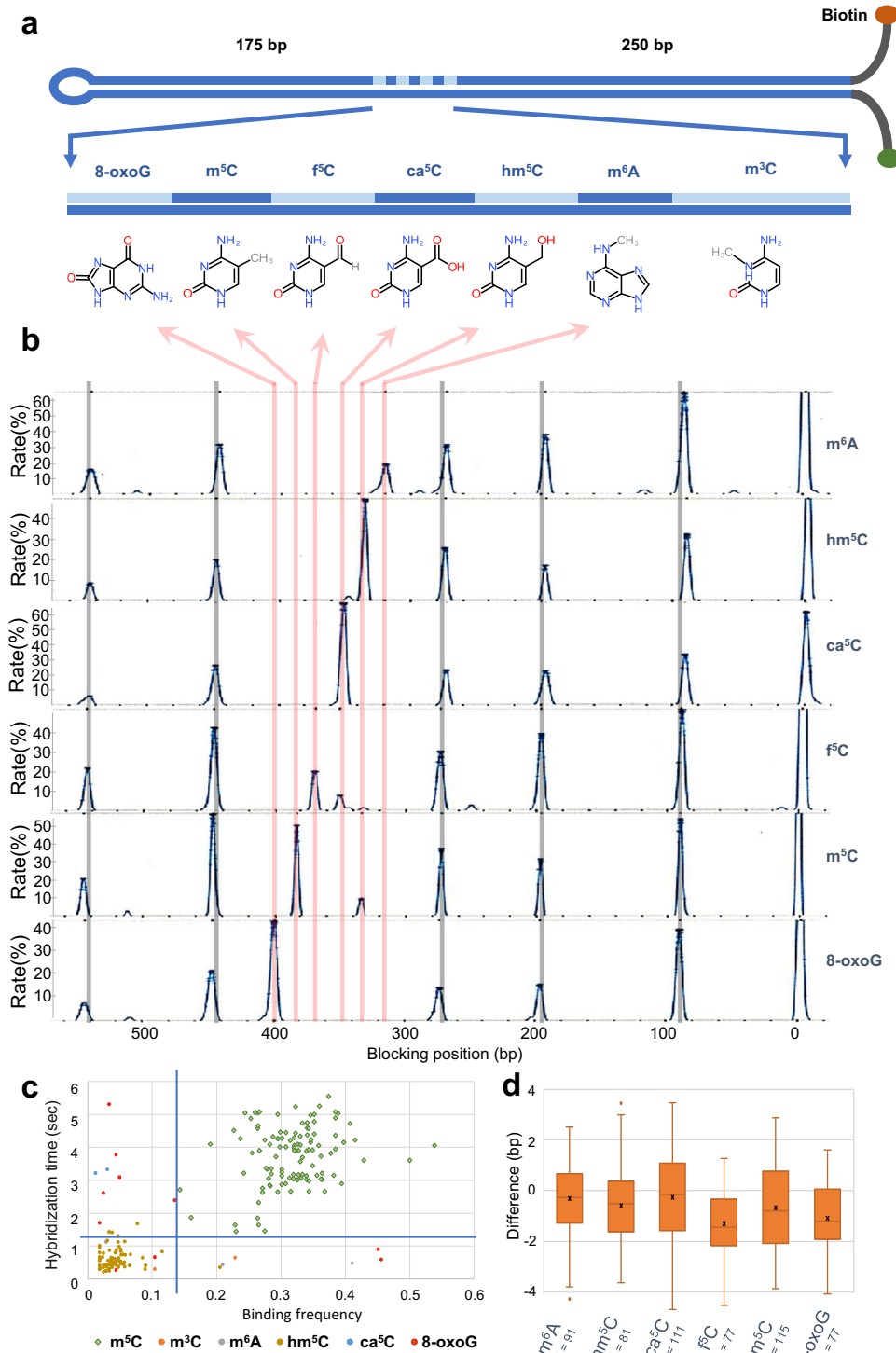

**Fig. 2 Detection and mapping of DNA base modifications. a** Schematic representation of the hairpin constructed with seven different base modifications. Each oligonucleotide contained a DNA base modification and the linker was constructed by annealing on a splint template. A detailed protocol how this hairpin is produced is available as Supplementary Fig. 1. **b** Detection of six out of the seven modifications present on the same molecule by sequentially injecting the antibody corresponding to each base modification. The six different experiments were aligned using the reference oligonucleotide bindings. **c** By plotting the hybridization time versus binding frequency, it is possible to cluster true positives (in this case, the m5C modification with the anti-m5C antibody) from the false positives (principally the hm5C modification). Each point represents the cumulative binding data for the m5C antibody as determined for each modified base on each individual hairpin (note that for many hairpins, there were no false binding events, so fewer points are plotted). By thresholding the time and frequency, we can eliminate the false positives. **d** Antibody binding positions were mapped to the hairpin molecule within 1 bp resolution for the majority of the molecules. Whisker boxes represent 50% of the points with the average as a line within the box and the median as a cross.

**Table 1 Sensitivity and specificity of DNA base detection with MTs using our antibody-based approach.**

| DNA-base modification | 8oxoG | m$^5$C | hm$^5$C | ca$^5$C | m$^6$A | f$^5$C |
|---|---|---|---|---|---|---|
| Sensitivity of detection | 100% | 98% | 95% | 98% | 100% | 100% |
| Specificity of detection | 100% | 100% | 98% | 100% | 100% | 100% |

Sensitivity is defined as the percentage of molecules in which the modification was successfully detected using the antibody among all the molecules observed, and specificity as the number of molecules where the modification corresponding to the antibody was correctly identified among all the blockages detected on each bead (either through thresholding or cross-reference between different antibodies).

improved the specificity of f$^5$C detection to 100% (Table 1). Lastly, for m$^3$C modification, we were unable to get repeatable data due to batch differences with the anti-m$^3$C polyclonal antisera that we tested (no commercial monoclonal antibodies were available for this modification).

**Characterization of splicing isoforms by oligonucleotide hybridization signature.** As our platform can analyze molecules of up to 5 kb with high precision, we explored its ability to identify and quantify full-length spliced isoforms. We used a well-studied mouse myogenic model[24], selecting two genes known to be alternatively spliced during mouse myogenesis: one with only two isoforms (*CAPZB*), and a second gene (*RBM9*), which has a more complex splicing pattern (Fig. 3a). Starting with full-length mRNA, we generated cDNA amplified using a linear amplification method and then incorporated these cDNAs into hairpins (Supplementary Fig. 5). Using a panel of oligonucleotides, we produced specific binding signatures that were dependent on the isoform (Fig. 3b). For *CAPZB*, we found that only 5% of the molecules present in myoblast cDNA included exon 8 whereas the number of molecules containing this exon increased to 65% in myotubes. These results were supported by splicing PCR analysis on independent samples from both cell types (Supplementary Fig. 6) and agree with the findings of Bland et al.[25]. For *RBM9*, four exons have been shown to be alternatively spliced giving rise to nine possible isoforms. We observed the expression of six of the nine possible isoforms, and four of these (isoforms 1, 2, 5, and 6) were enriched in only one cell type reflecting a change in their distribution upon muscle differentiation (Fig. 3a). To validate the measurement accuracy of our approach, we compared the expression levels of the different genes obtained by measurement of oligonucleotide hybridization in our instrument with read count data generated by the PacBio platform. For both *CAPZB* and *RBM9*, there was a close correlation in the measurement of the different isoform expressions (Fig. 3a and Supplementary Fig. 6).

**Decoding RNA with short oligonucleotide probes.** We next investigated whether we could decode an RNA template using a set of overlapping oligonucleotide probes and tested if short 3-mers could generate blockages of sufficient binding strength and duration to allow accurate positional information. We observed that modifying the oligonucleotide backbone structure with locked nucleic acid (LNA) and the incorporation of intercalators, such as acridine orange, improved their binding stability and allowed us to detect their binding with our platform.

A 100-base synthetic RNA template was ligated into a hairpin otherwise composed of DNA (same strategy as in Supplementary Fig. 4) and probed for blockage events with a set of 3-base modified oligonucleotides. Based on these blockage positions, we were able to reconstruct the sequence of the 100 bases of RNA. Each base was detected by three different oligonucleotides (as the binding positions overlap) which allowed for redundancy in the sequence determination and permitted a simple error-correction algorithm to generate the most energetically favorable sequence.

We were able to reconstruct sequence information from 20 of these molecules, obtaining sequence accuracies of between 70% and 96% for the individual 100-base RNA molecules compared to the expected sequence. The consensus sequence accuracy for these molecules was 95%, under the criterion that the correct base was called in the correct position in >50% of the molecules (Fig. 3c).

**Detecting epigenetic modifications in RNA.** We next analyzed the potential of using antibodies to identify and locate the position of different RNA base modifications in our platform. We constructed a hybrid DNA-RNA hairpin comprising 95-bases of synthetic RNA containing m$^5$C, m$^6$A, and inosine RNA base modifications ligated within a DNA backbone (Fig. 4a and Supplementary Fig. 4). The m$^5$C modification in RNA was detected with high sensitivity (on 99.1% of the molecules, $n = 111$, Fig. 4d). However, this antibody also generated blockages at base positions corresponding to the m$^6$A modification for 29 beads analyzed ($n = 111$ molecules). By comparing the binding profiles for different antibodies, we could filter out false positive peaks and thereby improve detection specificity. In the case of anti-m$^5$C antibody, we could eliminate blockages to m$^6$A by subtracting of the positions observed with anti-m$^6$A antibodies, increasing specificity to 98% (Fig. 4b, d).

For m$^6$A, by contrast, all three antibodies tested showed lower sensitivities (76–93%, Fig. 4d) than seen for m$^5$C detection. However, when combining the blockage data from all three m$^6$A antibodies, we saw a modest reduction in sensitivity (to 89.3%; Fig. 4b, d) but more importantly, raised the specificity of detection to 96%. Finally, the only commercially available inosine antibody we could identify, was not suitable for use in our assay due to high levels of cross reactivity against other base modifications and natural bases in the test template.

We also determined RNA base modification mapping precision by using a set of reference oligonucleotides targeting the DNA flanking the RNA region of the hairpin, to provide a series of calibration measurements. We observed a systemic difference of ~5 bp in the apparent position of the antibody blockage compared to the actual position of the RNA base modification, an effect observed for all antibodies tested. To investigate the cause of this bias, we hybridized a set of oligonucleotide probes to the RNA region containing the base modifications. These probes also showed bias in their calculated binding location, suggesting a differential force-extension response between the bases of RNA compared to DNA[26–28]. Comparing the binding positions between these internal oligonucleotides and antibody blockages allowed correction of this stretching effect and the accurate localization of the underlying base modification to within two bases (Fig. 4c).

**Targeting genomic loci in native DNA using a CRISPR-Cas enrichment protocol.** To efficiently analyze epigenetic modifications at defined loci within native DNA, we developed an amplification-free protocol for sequence-based enrichment. We chose the CRISPR-Cas meganuclease system as a starting point

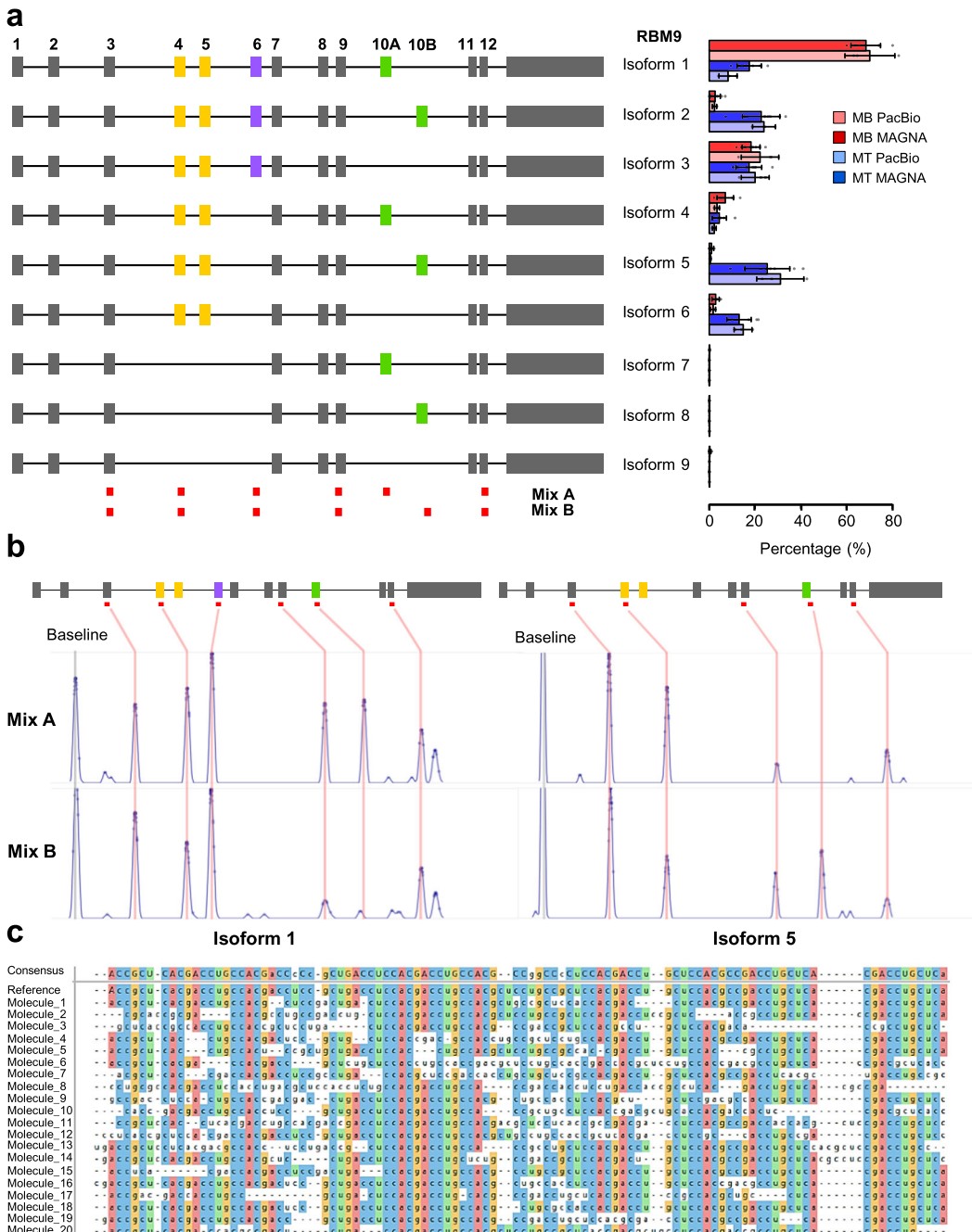

**Fig. 3 Identification of alternative splicing isoforms using fingerprinting oligonucleotides. a** A schematic showing nine different possible isoforms from mouse RBM9 transcripts. RNAs from two different cell types were used (myoblast, MB and myotube, MT), and different isoforms were quantified using the MAGNA platform or PacBio sequencing of long reads. The comparison between different identification methods, as well as changes in alternative splicing isoforms are shown on the right. **b** Examples of cDNA from two different isoforms generating specific signatures with the two oligonucleotide mixes to differentiate transcripts with either a 10A or 10B exon. **c** The alignment of reconstructed sequence from 20 individual molecules of RNA and the consensus sequence obtained from these sequences, where each base in the consensus is present in more than 50% of the molecules. The expected sequence (referred as reference) is indicated. Each base within a column that corresponds to the reference sequenced is colored. The consensus sequence showed 5 errors compared to the expected reference sequence.

based on the potential for high selectivity and compatibility with parallelization and minimal starting material requirements. We observed that both Cas9 and Cas12a proteins remain bound to their targeted DNA fragments even after cleavage, thus effectively shielding the cut site from exonuclease digestion (Supplementary Fig. 7). To exploit this property, we developed a two-step method where the region of interest was first protected by a pair of flanking CRISPR-targeted Cas12a proteins (Fig. 5a and a more

detailed schematic representation in Supplementary Fig. 8) and an initial enrichment was accomplished by digesting the non-targeted region using exonucleases. Then 3' overhangs were created using lambda exonuclease to digest from the ends of the fragment to a nested CRISPR-targeted dCas9 between 100 to 200 bases interior to the Cas12a sites. Finally, these ssDNA overhangs were used to assemble a hairpin by the ligation of Y-shape and loop sequences (Fig. 5a).

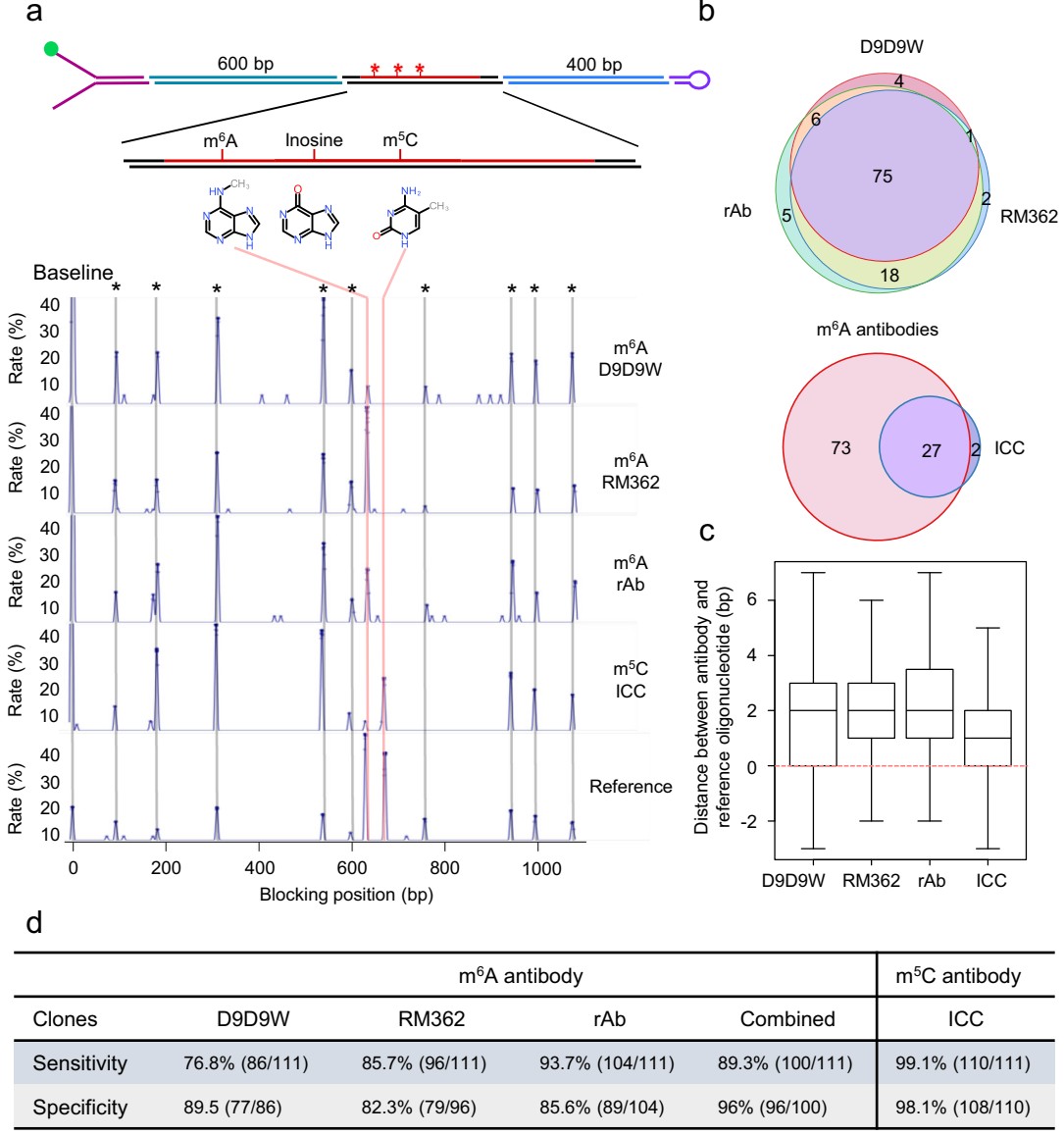

**Fig. 4 Detection and mapping of RNA base modifications. a** Representation of the synthetic hairpin constructed for epigenetic detection on RNA (see Supplementary Fig. 1 for more details). The red line represents RNA whereas black lines depict DNA. The Y shape adapter, to attach the hairpin on MyOne streptavidin bead via a biotin (green circle), as well as the loop are colored in purple. The binding position histogram of specific antibodies, as well as a reference oligonucleotide mix binding on a single hairpin is shown below. The reference oligonucleotides peaks are marked with an asterisk (*), and are used for alignment. The antibody binding positions are shown by the red lines. The last histogram represents the reference oligonucleotide binding positions that correspond to the same position as the modification. **b** Venn diagram showing the overlap between three different m6A antibody clones (top). Below, the Venn diagram shows the overlap between non-specific binding identified by m5C antibody (ICC) at the position of m6A modification, and the binding positions identified by at least two out of three m6A antibodies. **c** Boxplot showing the distance of binding between antibody and the reference oligonucleotides is base pair resolution for all antibodies tested. Whisker boxes represent 50% of the points with the average as a line within the box and the median as a cross. **d** Table showing the sensitivity and the specificity of all three m6A antibodies and the m5C antibody tested. For the combined analysis of m6A antibodies, a binding position was considered as such if it is recognized by at least two out of the three antibodies tested.

We quantified our approach and validated that it could retain epigenetic modifications by isolating four different DNA fragments from *E. coli* genomic DNA, ranging in size from 0.8 kb to 5 kb. Quantification by qPCR showed that we recovered between 55% and 75% of the starting material for the four fragments after the first step, and between 35% and 55% after the second step whereas the non-protected DNA decreased to less than 0.05% of the starting material (Fig. 5b). Most of the loss of targeted material can be accounted for by the two purification steps required during the protocol (almost 40% lost after the dCas9 step, Fig. 5b control without exonucleases). All four

fragments were successfully converted to hairpin molecules that could be analyzed on our platform (Fig. 5c).

We chose to study *E. coli* DNA because of the activity of the well-characterized *dam* and *dcm* methylases that modify A and C residues at well-defined sequence motifs, respectively. This allowed us to validate detection of both m6A and m5C on the same native DNA molecules via our antibody-based MT approach, two modifications not possible to map together using chemical-based techniques. First, we identified individual molecules using a single four-base oligonucleotide (CAAG) that bound multiple times to produce a characteristic 'genomic fingerprint'.

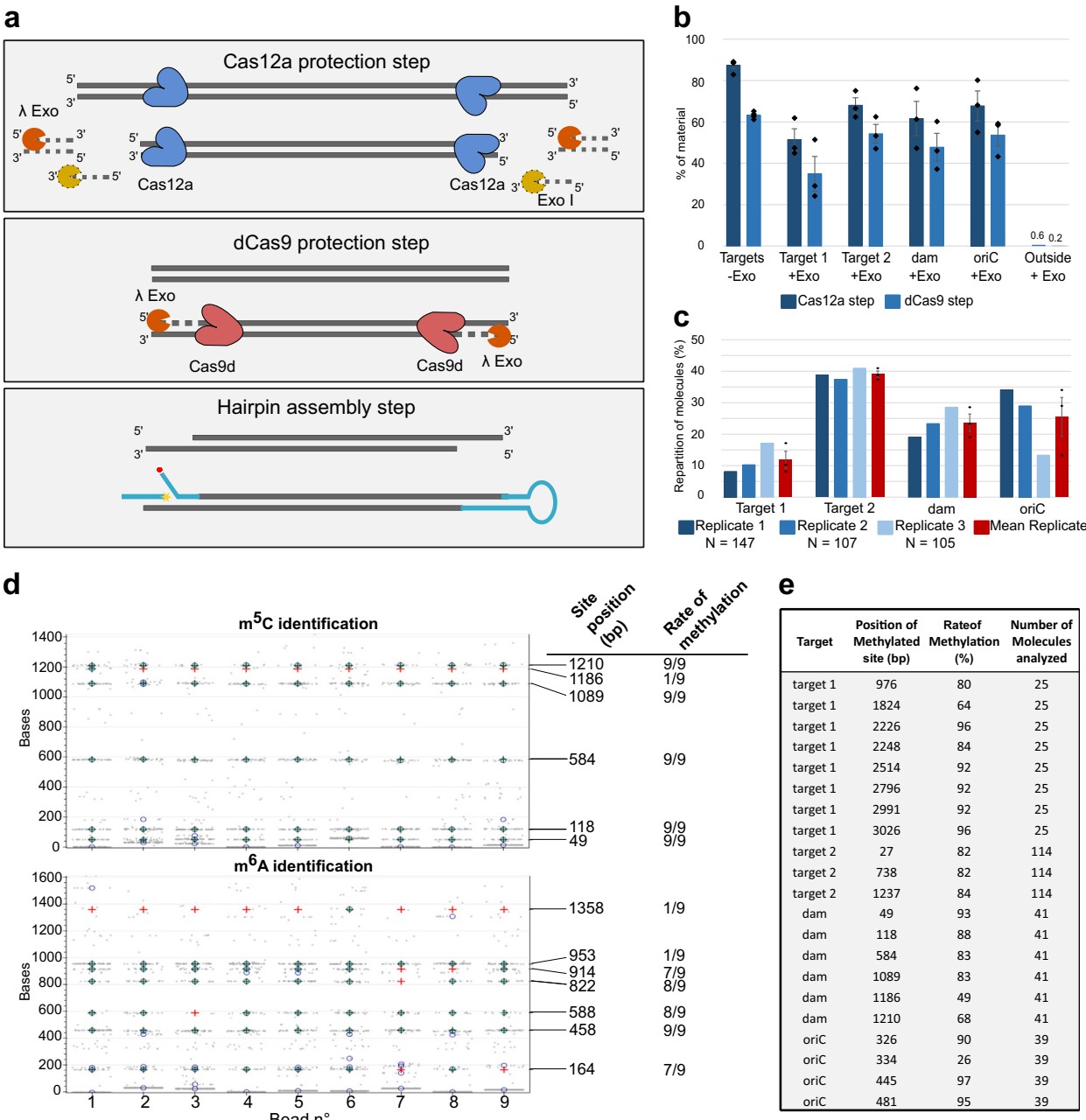

**Fig. 5 Enrichment of specific genomic regions using a CRISPR-based method. a** The enrichment protocol is divided into three major steps: (1) Flanking the region of interest with two Cas12a complexes, followed by digestion of all remaining genomic DNA with exonucleases. (2) Second, the generation of 3' ssDNA overhangs at each end of the fragment used to assemble the hairpins are created using the dead Cas9, followed by lambda exonuclease digestion. (3) Finally, this fragment is used to assemble the hairpin structure required for our platform. A more detailed protocol is available in Supplementary Fig. 8. **b** qPCR of enrichment of four *E. coli* targets. A control excluding the exonucleases was included (to account for purification loss) and a positive control for digestion was performed by quantifying off-target DNA. Protection was measured for each target after the Cas12a (dark blue) and dCas9 (light blue) steps. Bars represent the average protected material from three biological replicates +s.e.m, *n* = 3. **c** Repartition of target molecules analyzed on the MT platform from three biological replicates (shades of blue) and their average (red bars). **d** Detection of m5C and m6A on the same enriched *dam* molecules (each column represents a single molecule, and in each panel, the same column corresponds to the same bead). Gray points indicate detected binding events and the expected modification positions are indicated on the right axis. Blue crosses indicate detected blockages corresponding to the modification and red crosses indicate expected positions where methylation was not detected. **e** Analysis of m5C methylation of all the isolated *E. coli* fragments for all three biological replicates. The CCwGG site positions within the hairpins are indicated, as well as their rate of methylation.

Detection of such small oligonucleotides was again achieved by the same means as described for 3-mers above (characteristic binding times and hybridization rates are presented in Supplementary Fig. 9). All functional hairpins analyzed could be assigned to one of the four targets (*n* = 359 molecules),

demonstrating that the enrichment strategy was 100% specific (Fig. 5c). Next, we added the anti-m6A and anti-m5C antibodies sequentially to create blockages and then matched these to the expected locations of the *dam* and *dcm* recognition sequences (GAmTC and CCmWGG, respectively). All expected positions

were modified, albeit at different levels, typically over 50% and approaching 100% in some cases (Fig. 5d). However, there were some instances where we detected only a low level of methylation (e.g., m$^5$C at position 1186, and m$^6$A at positions 953 and 1358). We also assessed the level of m$^5$C methylation for all the isolated fragments (Fig. 5e) and observed systematic variations in levels of methylation dependent on the genomic position, consistent with data previously reported for exponentially growing cells[29].

**Enrichment of specific regions from human genomic DNA.** To demonstrate that our PCR-free enrichment protocol can also enrich human genomic fragments, we isolated four regions implicated in human diseases. We chose *FMR1* and *C9orf72* for their Short Tandem Repeat (STR) regions, which are difficult to measure on other platforms, and *SEPT9.1* and *SEPT9.2* which both have CpG islands, the methylation status of which has been implicated in colorectal cancer[30–32]. We performed the enrichment from cultured HEK cells and were able to isolate all four regions and confirm their identity using the specific blocking pattern produced by the oligonucleotide CAAG (see Supplementary Fig. 10 for examples of signature from these regions). As with our enrichment from *E. coli*, we did not observe any hairpins containing off-target DNA. Therefore, our enrichment worked on human gDNA, as well as on *E. coli* with 100% target specificity.

**Analysis of STR length and methylation status at the *FMR1* locus.** To further demonstrate the ability of our platform to extract both genetic and epigenetic information on the same individual molecules, we focused on Fragile X Syndrome (FXS). This genetic disorder is characterized by an expansion in the number of CGG repeats in the 5' untranslated region (5'UTR) of the Fragile X mental retardation 1 (FMR1) gene on the X chromosome[33]. In addition to the repeat expansion, the FMR1 promoter can also be differentially methylated in different disease states[34]. We tested whether our workflow could provide an accurate method to measure the lengths of the trinucleotide repeats and, by preserving epigenetic marks during sample preparation, allow the analysis of both features together in the same individual DNA molecules.

In a pilot experiment with genomic DNA isolated from the human HEK cell line, we found that the CCG repeat sequence created additional spontaneous blockage events, most likely due to the formation of DNA secondary structures. To overcome this effect, we designed new reference oligonucleotides that could form a three-way junction on either side of the repeated region, thereby transiently preventing the hairpin from opening under high force (such that any secondary structures are unfolded before the unzipping occurs (Supplementary Fig. 11). Using this strategy, we were able to measure the distance between blockings in the hairpin opening phase rather than in the rezipping phase. This approach was validated by quantifying a normal range of trinucleotide repeats in HEK cells (between 31 and 35, $n = 11$) (Fig. 6a, c).

We then used our CRISPR-Cas-based enrichment protocol to target the FMR1 locus from cultured cells of sample NA06896, derived from an unaffected female heterozygous carrier of FXS, and measured repeat lengths using this opening assay. We observed that approximately half of the individual DNA molecules had a repeat structure characteristic of a normal allele (between 21 and 28 repeats, $n = 30$) and the others carried an expanded repeat count (>50 repeats, $n = 22$, Fig. 6a, c). These data matched the expected ratio for a heterozygous sample. As expected from these repeats, we observed a greater variability among the molecules with an expanded repeat number compared to the normal allele. Twenty percent of molecules had a repeat

number greater than 200, the threshold for the full mutation form of FXS. The other molecules with expanded repeats fell in the range defined for pre-mutation carriers.

Next, we analyzed the methylation status on the same single molecules from NA06896 with an antibody against m$^5$C. For all molecules with over 200 repeats, we observed either no or very low levels of methylation in the promoter region ($n = 3$, Fig. 6b). This result was striking in comparison to most molecules in the normal or pre-mutation category which had high levels of methylation at both CpG and non-CpG sites. However, even among these two groups, we did find instances of low methylation, but they were less frequent than the highly methylated loci (Fig. 6b). Our results were in concordance with Chen et al.[35] who used a methylation-specific PCR test and reported a lack of methylation on repeats greater than 150, but high levels of methylation for pre-mutation alleles and low levels for normal repeats[35]. Our amplification-free enrichment method was effective for targeting FMR1 and for preserving the underlying genetic and epigenetic structure of the locus.

## Discussion

Over recent years, it has become increasingly apparent that there are areas of genome biology that are difficult to access with NGS, such as the accurate detection of base modifications in DNA and RNA, the co-detection of multiple genomic features on the same molecules, and obtaining long-range genomic information[36]. These areas of biology have fundamental importance to our understanding of how genomes are organized and controlled, and our ability to characterize the different processes leading to disease[37]. There is therefore a need for improved genomics tools that can overcome these shortcomings of NGS.

We have presented major advancements in the MT technology outlined in Ding et al.[22]. First, we demonstrated direct detection of sequence signatures and modified bases together on both DNA and RNA molecules and their localization at single base pair resolution. Second, to capture this diversity in native DNA, we have developed a highly specific amplification-free workflow for enriching specific genomic regions of various sizes from complex prokaryotic and eukaryotic genomes. The non-destructive nature of the MT technology—we have opened and closed nucleic-acid hairpins up to 10,000 times—means that a wide variety of probes can be used to both detect multiple features and improve the specificity and sensitivity of sequence and base modification detection at the single molecule level.

NGS-based methods cannot detect these base changes directly and instead typically rely on the chemical conversion of one base to another[38,39]. Due to the subtractive nature of the analysis, maps are generated for only one modified base at a time. By contrast, single molecule approaches offer the promise of a direct readout of unamplified native DNA changes. There has been steady but slow progress in the detection of base modifications using both nanopores (Oxford Nanopore, ONT) and Single Molecule Real Time sequencing (SMRT, Pacific Biosciences)[40–44]. Detecting modified bases with SMRT sequencing requires a minimum sequencing fold coverage of between 25× (e.g., for m$^6$A) and 250× (e.g., for m$^5$C)[45], while with ONT protocols the single pass of any individual template through a pore provides only one opportunity to detect modified bases in that molecule. Unlike these other single molecule platforms, our MT approach has allowed the unambiguous identification of six different modified bases in the same DNA template, to our knowledge the largest variety of modified bases that have been detected in an individual template with high accuracy using any genomics approach.

Highly accurate detection of multiple base modifications on MTs is achieved by repeated probing of the same templates with

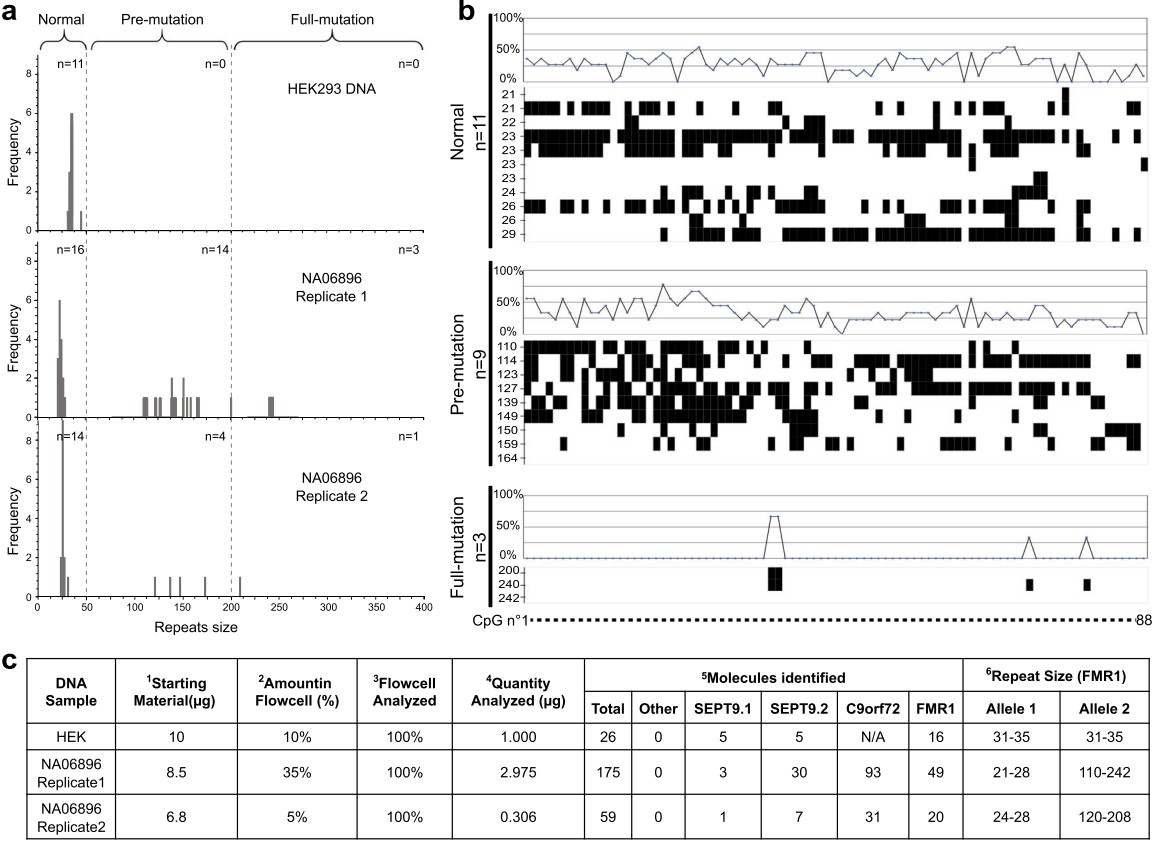

**Fig. 6 Analysis of repeat length and methylation status of the FMR1 locus. a** Histograms of the distribution of CGG repeat length for FMR1 on two DNA samples (HEK DNA, and the clinical sample, NA06896). The n represents the number of molecules identified in three categories, normal (<50 repeats), pre-mutation (50 to 200 repeats) and full-mutation (>200 repeats). **b** Cytosine methylation analysis of the CpG island located within the FMR1 promotor region of DNA sample NA06896. All CpG or non-CpG sites are represented on the X axis (the list and position of sites are in Supplementary Table 1). Molecules are ordered by repeat size and grouped by mutation status (normal, pre-mutation, full-mutation). Line graphs represent the frequency of molecules identified as methylated for a specific CpG or non-CpG site within this population. **c** The table summarizing the libraries prepared and the results obtained from these samples. (1) Amount of DNA used to prepare the library. (2) Amount of starting material injected in the flow cell. (3) Proportion of the flow cell analyzed. (4) Relative quantity of starting material analyzed. (5) Number of molecules analyzed on the MT platform and repartition according to the oligonucleotide binding pattern. (6) Quantification of FMR1 repeat size, N/A: Not included in the library preparation.

different antibody probes and analysis of their binding kinetics. These features overcome both the lack of specificity that has been well documented for ChIPseq-type experiments that use only a single antibody[46], and the inherent stochastic sampling that contributes to the high error rates often observed in other single molecule approaches. Indeed, using our approach, we have been able to clearly show the cross-reactivity profiles of a number of commercial antibodies used routinely in epigenetic analysis. We anticipate that the standardization of these antibody-based tools and the generation of new appropriately tuned ligands based on proteins that naturally bind to modified bases will greatly assist in the characterization of the landscape of epigenetic changes and damaged bases on nucleic acids.

Over 150 base modifications are found in RNA and their functions are only just being discovered as it remains a technical challenge to identify the many similar chemical moieties at base-pair resolution[36,47]. As with DNA, single molecule techniques offer benefits over current NGS-type approaches in reading RNA modifications directly in their native strands, rather than indirectly through immunoprecipitation or reverse transcriptase stuttering[48]. We confirmed that analytical approaches we developed for DNA could also be applied to RNA. Antibodies were used to identify and localize two biologically important base modifications present on the same individual RNA molecules with high sensitivity and specificity.

We intend to expand this approach to wider range of RNA base modifications, helping to unlock this important emerging area of biological science.

To analyze base modifications and genomic variation of specific genomic loci, we developed a CRISPR/Cas enrichment method that can isolate native, high molecular-weight fragments of DNA from complex genomes with exceptionally high target specificity (in this paper we demonstrate 100% on-target rates). This high specificity comes from the combination of three highly specific steps in our protocol: the Cas12a step, dCas9 step, and finally the use of specific oligonucleotides to assemble the hairpins. Even if there are off-targets for these Cas proteins, a region must be flanked by two closely spaced off-targets to be protected from degradation, an unlikely occurrence especially when few targets are enriched.

The method has the benefit of not being affected by genetic variability within target regions. Furthermore, it is not only compatible with our hairpin-based MT assays but, by ligating alternative adapters, can also be used with both NGS and other single-molecule sequencing platforms. We envisage it will be applied to the analysis of specific disease-related genetic regions where sequence and epigenetic variability play important pathogenic roles. Also, the high specificity of the approach may help to reduce the need for costly deep sequencing and see it brought into diagnostic pipelines.

To help demonstrate its utility, we used FMR1 as an example of a clinically relevant gene that is difficult to analyze with short-read technology because the number of trinucleotide repeats required for diagnosis exceeds typical NGS read-lengths, as well as its 100% GC content, preventing good amplification during NGS sample preparation. Moreover, we were interested in studying FMR1 at a single-molecule level because of the link between the epigenetic status of the FMR1 promoter and the number of CCG repeats, and since repeat number and methylation mosaicism are common phenomena in FXS. From human genomic DNA, we enriched fragments containing FMR1 and observed that the allele frequency of expanded repeats to normal was in the expected 1:1 ratio for a heterozygous sample. We were able to accurately measure the length of trinucleotide repeats in normal alleles and those in the full-mutation state, as well as identify DNA molecules in a pre-mutation status. In addition, the same single molecules were interrogated for promoter methylation, and we identified that all molecules that had >200 repeats were almost entirely unmethylated. These data are in broad agreement with Chen et al.[35] using orthogonal methods on the same sample[35]. Analysis of these genetic and epigenetic features on native molecules of FMR1 has been attempted using both SMRT sequencing and nanopore sequencing but in neither case was it possible to perform accurate repeat sizing and methylation analysis on the same single molecules[49,50]. Our results suggest that the combination of our highly specific targeted enrichment together with our MT analysis provides a fast and accurate workflow for the detection of multilayered patterns of information, allowing comprehensive epigenetic profiling in clinical samples from a single assay.

In conclusion, we have shown how a MT platform can be used to perform a variety of genomic and epigenetic analyses on kilobase-length single molecules of both DNA and RNA and that these analyses can be performed sequentially to reveal multiple features from the same molecules. Furthermore, as MTs are already used for characterizing protein/nucleic acid interactions, we see the utility of this technology to develop a powerful multiomic analysis platform with which information on sequence variation and epigenetic modification is combined with that from the binding and modification of protein complexes that regulate gene expression. To this end, we are also planning to integrate fluorescence detection into the MT technology, offering a broad platform for the analysis of proteins and nucleic acid interactions[51].

Efforts are also underway to scale the technology from current throughputs, where hundreds of molecules are analyzable in parallel, to millions of molecules. This will be achieved using both optical and all-electronic methods (patent EP3090803B1) and will allow the technology to be used either for genome-wide and transcriptome-wide analyses or, alternatively, when in combination with our amplification-free enrichment method, for fast and cost-effective panel-based testing for disease-causing genes at high depths of coverage. Finally, we intend to reduce the required starting material for library construction to levels that will open up the areas of single-cell genomic, epigenomic, and (epi)transcriptomic analysis.

## Methods

### Hairpin construction and characterization of the new magnetic tweezer instrument

For measuring the precision of the new instrument, the DNA epigenetic hairpin (600 bp) described in the next section was prepared and attached to paramagnetic MyOne T1 streptavidin beads (Dynal/Life Tech).

To measure Brownian noise on the new SDI instrument, a hairpin was constructed by amplifying a 5 kb fragment from the PhiX174 genome (Supp Fig. 1). Each primer used (see Supplementary Data 1) contained a non-palindromic restriction site (BsaI), which introduced two distinct 4-base overhangs at each end of the PCR fragment after digestion. These 4-base overhangs allowed the

directional cloning of the Y-shaped adapter required for bead and surface attachment (Triple-biotin/PS866), as well as the synthetic loop (PS359) at the other extremity (Supplementary Data 1). The resulting hairpins were bound to MyOne™ T1 streptavidin beads (Dynal/Life tech) and injected into a flow cell for capture through a splint oligonucleotide (PS867) on the surface. This was achieved by first covalently attaching an oligonucleotide (PS625) containing a DBCO group at the 3' end to an azide coated coverslip (PolyAn, GmbH) using copper-free click chemistry that would form the base of the flow cell. Eleven-base reference oligonucleotide in ABB6 buffer (20 mM Tris HCl pH 7.5, 150 mM NaCl, 2 mg/mL BSA, 0.1% sodium azide) was injected into the flow cell and test cycles (of increasing and decreasing magnetic force) were performed at 18 °C. The noise variation between successive camera frames (frequency 1/30 s.) was extracted and plotted for all the oligonucleotide binding positions on each hairpin.

### Hairpin construction for DNA epigenetic detection

Seven synthetic oligonucleotides, each containing a DNA base modification, were obtained from Eurogentec along with the complementary oligonucleotide that allows construction of the synthetic linker (Supplementary Data 1). A synthetic 175 bp PCR fragment was digested with BsaI to generate a non-palindromic 4-base overhang and ligated to the synthetic linker. A second PCR fragment of 250 bp, also digested with BsaI to generate a different compatible 4-base overhang, was ligated at the second extremity of the synthetic fragment. The resulting fragment was digested with BsmBI to generate third and fourth non-palindromic sites at either end of the 550 bp fragment to allow the ligation of the Y-shape (triple-biotin/PS866) and the synthetic loop (PS359) (Supplementary Fig. 1). The resulting hairpin was purified on an agarose gel and attached to paramagnetic Dynabeads™ MyOne™ streptavidin T1 beads (Invitrogen).

### Detection of base modifications in DNA

For base modification detection, commercial antibodies raised against the DNA base modifications present on the test hairpin were used. For m5C, we used the ICC/IF clone from Diagenode (C15200003) at a dilution of 1/500. For m6A, we used the Cell Signaling Technology clone D9D9W at a dilution of 1/300. For 8-oxoG detection, we used the clone 15A3 from R&D Systems Europe Limited (4354-MC-050) at a dilution of 1/500. For hm5C, we used the clone RM236 from Invitrogen™ (15815913) at a dilution of 1/500. For ca5C detection, we used the clone RM24 1-A3 from AbCam at a dilution of 1/500. For f5C, we used the polyclonal antibody mix from Active Motif® (61228) at a dilution of 1/500. These antibodies were diluted in ABB6 buffer and supplemented with 500 nM of the reference oligonucleotide OR3 for precise mapping of the modification of the hairpin. At least 100 cycles of opening-closing were performed for each antibody and the binding of the oligonucleotide, as well as the antibodies were aligned to the known sequence of the hairpin.

### Hairpin construction for RNA epigenetic detection

A synthetic hairpin was constructed from synthetic RNA fragment flanked by 600 bp and 400 bp DNA fragments. In brief, the three RNA oligonucleotides (Eurogentec) containing three different base modifications: m6A, inosine and m5C, separated by 20 nucleotides, were assembled by hybridization and ligation over a complementary DNA strand using T4 RNA ligase 2 (NEB). To construct the hairpins, the resulting synthetic RNA/DNA hybrid fragment with DNA overhang was ligated to both the 600 and 400 bp DNA fragments before ligating a loop (PS359) and Y-shape (triple-biotin/PS866) at either end using T3 DNA ligase (NEB®) (strategy similar to Supplementary Fig. 1, except that the PCR fragments used were different). The hairpins were gel purified and attached to paramagnetic Dynabeads™ MyOne™ streptavidin T1 beads (Invitrogen).

### Epigenetic detection of RNA using antibodies

To detect epigenetic base modifications on the synthetic RNA-containing hairpins, several commercially available antibodies were used. For m5C, the ICC/IF (C15200003) mouse monoclonal antibody from Diagenode was used at 1/250. For m6A, we used the rabbit monoclonal m6A antibodies from Cell Signaling Technology (clone D9D9W, at 1/500 dilution), from RevMAb Biosciences (clone RM362, at 1/300 dilution) and the mouse monoclonal recombinant AbFlex m6A antibody (rAb) from Active Motif (at 1/300 dilution). A mixture of reference DNA oligonucleotides that correspond to the position of the modified nucleotides was used to compare with antibody binding (the last base at the 5' end of the oligonucleotide hybridizes with the modified base, blocking the fork in the same position as the antibody). Each experiment was recorded for more than 100 cycles. For all the experiments, the antibody was used together with reference oligonucleotides in oligonucleotide binding buffer (PBS pH 7.4, 2 mg/mL BSA, 0.1% sodium azide).

### mRNA isolation and cDNA library preparation

C2C12 myoblast and myotube cells were gifts from Dr. Herve Le Hir's laboratory, from the biology department at ENS, Paris, France. Total RNA was extracted with Trizol (ThermoFisher) following the manufacturer's instructions and treated with TurboDNAse (Ambion).

The full-length cDNA was synthesized from 2 µg of total RNA using the TeloPrime Full-Length cDNA Amplification Kit (Lexogen) and the final full-length cDNA library was amplified 15–20 cycles according to the manufacturer's instructions.

**Alternative splicing PCR**. We performed classical PCR for splicing isoform detection for CAPZB and RBM9 using primers on the flanking constitutive exons. PCR was performed using DreamTaq (Fermentas) for 30 cycles of denaturation at 95 °C for 60 s, followed by a step at 60 °C for 30 s. and elongation at 72 °C for 4 min. The resulting PCR sample was loaded onto a fragment analyzer (Agilent Technology) and the ratios between different peaks were calculated using the area under the curves. A list of the primers used is given in Supplementary Data 1.

**Loop PCR and hairpin preparation**. Loop PCR was performed using a forward primer containing either an isoG base (to stop the polymerase and therefore create a 5' overhang of known sequence) or a nickase site, and a reverse primer containing a loop sequence (the sequence of the oligonucleotides used can be found in Supplementary Data 1). PCR was performed with Phusion DNA polymerase (NEB®) and we used a slow ramp for the elongation step from 72 °C to 98 °C to allow the synthesis of the loop on the reverse primer. After 20 to 40 cycles of PCR, depending on the expression of the target, exonuclease I (ExoI, NEB®) was added to remove the single-stranded DNA bearing a 5' loop, as well as the remaining primers. Two more cycles were performed to fill in DNA molecules with a 3' loop, and a second ExoI step was used before the PCR products were purified with columns (Qiagen or NEB®) or Kapa beads (Roche Diagnostics). When isoG forward primer was used, a Y-shape containing an isoC overhang was ligated using T4 DNA ligase (Enzymatics), followed by agarose gel purification. In the case of the nickase site, PCR fragments were treated with Nb.BbvCI nickase (NEB®) before purification and ligating the Y-shape with the correct overhang. The final hairpins were attached to Dynabeads™ MyOne™ streptavidin T1 beads (Invitrogen).

**Quantification of splicing isoform hairpins by magnetic tweezer analysis**. For the analysis of splicing isoforms, a set of oligonucleotides were designed for the two genes studied that hybridized to both constitutive exons and alternatively spliced exons (Supplementary Data 1). The concentration of each oligonucleotide was optimized such that their binding rates (proportion of cycles for which blockage occurred) were similar. The alternative splicing isoforms were determined by the presence or absence of the oligonucleotides binding to the alternative exons. We performed three technical replicates (hairpin construction from the same full-length cDNA library) for three biological replicates (RNA from different cell passages). For each gene, more than 100 beads were recorded, and the identities of the isoforms were determined from hybridization patterns. The proportion of each isoform was calculated for all beads analyzed per technical replicate and represented as the average of all three biological replicates.

**PacBio sequencing**. For SMRT sequencing, we used the same reverse loop primer and a forward primer integrating a four-base barcode that allows multiplexing of different samples. PCR was performed using Phusion DNA polymerase, with HF buffer for CAPZB, and GC buffer for RBM9. The resulting PCR fragments were purified on agarose gel and multiplexed before SMRT sequencing (Eurofins Genomic).

To improve the assignment rate, we took the unassigned reads and re-mapped them specifically to the two genes with all the isoforms using Burrow Wheeler Aligner (BWA). The resulting BAM files were converted to SAM files and custom scripts were used to re-assign the mapped sequences to correct samples according to the barcodes. The CIGAR tag in SAM files for each sequence was used, and any reads not mapping from the beginning of the sequences were discarded. In addition, any reads that did not map to either strand were also discarded. The resulting sequences were searched for barcode at the beginning or end of the reads depending on the mapping orientation, and any chimeric reads were cut and searched for barcodes. If both parts of the chimeric sequences could be correctly assigned, the reads were split into the correct demultiplexed sample files. If the barcode was too short, the reads were discarded, and if the barcode was wrong for the mapped sequence, these reads were discarded but counted towards the 'mismatch' number. In the end, all the correct sequences were pooled for each sample and output as a BED file.

**Magnetic tweezer analysis of RNA using short oligonucleotide probes**. For decoding of RNA using short oligonucleotides, a synthetic fragment of 100 nucleotide 2'-O-Me RNA and its complementary DNA with DNA overhangs was ligated to 170 bp and 250 bp DNA fragments at either side using T3 DNA ligase (Enzymatics). The resulting sequences were ligated to a loop (PS359) and Y-shape adapter (triple-biotin/PS866) using T3 DNA ligase (Strategy similar to Supplementary Fig. 1 except that the RNA strand did not contain modification). The hairpins were agarose gel purified and attached to MyOne™ T1 streptavidin beads (Dynal/Life Tech). The attached hairpins were then ligated to the oligonucleotide at the surface of the flow cell using T3 DNA ligase for 30 min. After the ligation step, un-ligated molecules were washed using 20 mM NaOH followed by neutralization with 50 mM Tris-HCl. The methylation group of 2'-O-Me RNA bases protects the ribose from hydrolysis, allowing the washing step with NaOH. However, this washing step is not performed when hairpins containing natural RNA bases are used.

After defining the optimal force to open and close the hairpins, we injected the oligonucleotides into the flow cell sequentially and recorded at least a hundred cycles. Each 3-base oligonucleotide was tested individually at a concentration varying between 0.5 and 5 nM along with reference oligonucleotides (at 150 nM). Blocking positions and time of blockages were extracted for each cycle for each bead and analyzed using our custom developed software.

**Enrichment using Cas proteins**. *E. coli* genomic DNA was extracted using Quick DNA Plus kit from liquid culture of NEB® 5-alpha Competent *E. coli* (NEB #C2988) grown in LB medium according to the protocol provided by the manufacturer (Zymo Research). Purified NA06896 DNA was obtained from the Coriell Institute.

CRISPR-Cas12a (Alt-R® CRISPR-Cpf1) and CRISPR-dCas9 (Alt-R® S.p. dCas9) were purchased from IDT. All crRNA and tracrRNA guide RNA, either for Cas12a or Cas9, were chemically synthesized by IDT. For each target, two Cas9-crRNA were designed to flank the region of interest to be protected and two Cas12a-crRNA at least 100 bases away from the Cas9-crRNA position. (The complete list of crRNA is available in Supplementary Data 1).

For the Cas9 gRNA, each crRNA–tracrRNA duplex was prepared independently by mixing 1 pmole of tracrRNA with 2 pmoles of crRNA in Nuclease Free Duplex Buffer (IDT), heated at 95 °C for 5 min., then cooled at in successive steps at 80 °C for 10 min., 50 °C for 10 min., and then 37 °C for 10 min. Each Cas9 protein was loaded with a gRNA complex separately in NEB3.1 Buffer (100 mM NaCl, 50 mM Tris-HCl, 10 mM MgCl₂, 100 µg/ml BSA, pH 7.9) at 25 °C for 15 min at room temperature at a final concentration of 500 nM for dCas9 and 1 µM for the RNA guide.

crRNA guides for Cas12a were folded in NEB2.1 buffer (50 mM NaCl, 10 mM Tris-HCl, 10 mM MgCl₂, 100 µg/ml BSA) at 80 °C for 10 min., followed by a step at 50 °C for 10 min, then 37 °C for 10 min. Each Cas12a/crRNA complex was assembled in NEB2.1 buffer supplemented with 10 mM DTT for 15 min at room temperature at a final concentration of 500 nM of Cas12a and 1 µM crRNA.

*Enrichment of E. coli targets*. 2.6 pmol of each Cas12a/crRNA complex (2 complexes per targets) was mixed with 5 µg of genomic DNA in NEB2.1 Buffer supplemented with 10 mM DTT for 90 min at 37 °C. A mixture of exonucleases (lambda exonuclease (40 U/µg of gDNA), ExoI, (40 U/µg of gDNA)) was added to the reaction and incubated for an additional 90 min at 37 °C. Inactivation of the reaction was performed using 40 ng Proteinase K and EDTA at a final concentration of 20 nM, followed by a purification using 0.8× KAPA Pure beads (Roche). The DNA was repaired by T4 DNA polymerase (NEB®) with dNTP (200 µM) in NEB3.1 buffer to repair the 5' overhang for 15 min at 12 °C followed by a purification step using 0.8× KAPA Pure beads. The resulting DNA was incubated with 333 fmol of dCas9/gRNA complex in CutSmart Buffer supplemented with 0.1% Triton X-100 for 60 min at 37 °C. Lambda exonuclease (20 U/µg of gDNA) was added to the reaction and incubated for an extra 90 min at 37 °C. The reaction was inactivated using 40 ng Proteinase K and EDTA at a final concentration of 20 nM, followed by purification using KAPA Pure beads (1×). The enriched fragments, which contained a long 3' ssDNA overhang, were incubated with 1 pmol of target specific biotin, surface and loop oligonucleotide (sequence available in Supplementary Data 1), 40 U Taq DNA Ligase (Enzymatics), 0.25 U of Bst DNA Polymerase Full Length (NEB) in ThermoPol® Reaction Buffer (NEB®) supplemented with 200 µM dNTP, 1 mM NAD⁺ at 50 °C for 30 min. Excess oligonucleotide was then digested with 25 U ExoI for 30 min at 37 °C, followed by a purification using KAPA Pure beads (1×). The DNA fragments were digested with BsaI to generate specific overhangs that allow the ligation of surface specific oligonucleotides (PS1420 and PS867) and the second overhang created at the opposite end was used to ligate the loop (PS421). The resulting hairpin molecules were bound on 5 µl of Dynabeads™ MyOne™ T1 Streptavidin (Dynal/Life Tech) in passivation buffer (PB: PBS pH 7.4, 1 mM EDTA, 2 mg/mL BSA, 2 mg/mL pluronic surfactant, 0.6 mg/mL sodium azide) for 60 min at room temperature. Beads were washed and resuspended in 1× PB prior to loading into the flow cell.

*Quantitative PCR of enriched fragments from E. coli*. The efficiency of *E. coli* target enrichment was quantified by qPCR after both protection steps. For each target, we designed primer pairs between the two dCas9 and a pair of oligonucleotides outside the targets as a negative control. qPCR reactions were performed on a QuantStudio instrument (Applied Biosystems) with the Fast SYBR™ Green Master Mix Kit (Applied Biosystems).

The enrichment efficiency was calculated based on three biological replicates and each qPCR reaction was performed in triplicate. Amplification reactions were performed in final volumes of 20 µL consisting of 1× Fast SYBR™ Green Master Mix, 50 nM of each forward and reverse primer and 8 µL of DNA template (1:700 dilution of the enrichment reaction). The qPCR cycling conditions were as follows: initial denaturation at 95 °C for 10 min., followed by 40 cycles of denaturation at 95 °C for 15 sec., annealing and elongation at 60 °C for 1 min. Melting curves were produced by increasing the reaction temperature from 60 °C to 95 °C. Standard curves were performed using known amounts of *E. coli* genomic DNA. The initial amount of genomic DNA was set as 100% and the efficiency of each protection step was calculated by dividing the quantity of remaining DNA

after exonuclease protection by the initial amount of genomic DNA. Two controls were included: (1) the same protocol for all the four targets was performed but without the exonucleases, allowing estimation of the percentage of material lost during the purification; (2) a region outside the four targets was quantified to determine the amount of remaining material not protected after exonuclease digestion.

*Enrichment of FMR1 on human genomic DNA.* 300 fmol of Cas12a/crRNA complex per µg of gDNA were incubated in NEBuffer™ 2.1 buffer supplemented with 10 mM DTT for 60 min at 37 °C. A mixture of exonucleases (lambda exonuclease, 20 U/µg of gDNA and ExoI, 20 U/µg of gDNA) was added and the reaction was incubated for another 60 min at 37 °C. Inactivation of the reaction was performed using 40 ng Proteinase K and EDTA at a final concentration of 20 nM, followed by a purification using KAPA Pure beads (1×). The 5' overhang was repaired using T4 DNA polymerase (NEB®) with dNTP (200 µM) in NEBuffer™ 3.1 buffer for 15 min at 12 °C followed by a purification step using 0.8× KAPA Pure beads. The resulting DNA was incubated with 150 fmol of dCas9/gRNA complex per µg of initial gDNA in NEBuffer™ 3.1 buffer for 60 min at 37 °C. Lambda exonuclease (15 U/µg of gDNA) was added and the reaction incubated for another 60 min at 37 °C. Inactivation of the reaction was performed using 40 ng proteinase K and EDTA at a final concentration of 20 nM, followed by a purification using KAPA Pure beads (0.8×). The enriched fragments, which contained a long 3' ssDNA overhang, were incubated with 1 pmol of target specific biotin, surface and loop oligonucleotide (sequence available in Supplementary Data 1), 40 U Taq DNA Ligase (Enzymatics), 0.25 U Bst DNA Polymerase Full Length (NEB®) in ThermoPol® Reaction Buffer (NEB®) supplemented with 200 µM dNTP, 1 mM NAD+ at 50 °C for 30 min. Excess oligonucleotide was then digested with 25 U ExoI for 30 min at 37 °C, followed by a purification using KAPA Pure beads (1×). The DNA fragments were digested with BsaI to generate specific overhangs that allow the ligation of surface specific oligonucleotides (PS1420 and PS867) and the second overhang created at the opposite end was used to ligate the loop (PS189 and PS1472). The resulting hairpin molecules were bound on 5 µl of Dynabeads™ MyOne™ Streptavidin T1 (Invitrogen) in PB for 60 min at room temperature. Beads were washed and resuspended in PB prior to loading into the flow cell.

**Determination of FMR1 repeat length and promoter methylation using magnetic tweezers.** The prepared hairpin-beads were injected in a flow cell and the small tandem repeat sizes were analyzed using our opening assay with ten oligonucleotides capable of forming a three-way junction, which were targeted to hybridize specifically to invariable sequences located downstream and upstream from the CGG-repeat position (Supplementary Fig. 11). These special oligonucleotides capable of forming a three-way junction during the opening phase contain a single-stranded region of 8 to 10 bases depending on the sequence, as well as a loop structure complementary to the sequence located upstream of the blocking position. Upon binding, these oligonucleotides cause a transient blockage in the opening of the hairpin, allowing the precise mapping of the position of blockages. The number of repeats was then determined by constructing a histogram of blocking positions and aligning the two flanking constant regions to precisely determine the number of repeats on the variable region. For base modification detection, the anti-m5C antibody clone ICC/IF (Diagenode) was added to the flow cell at a 1:500 dilution in ABB6 buffer.

**Statistics and reproducibility.** All the mapping positions of antibodies and oligonucleotides presented in Figs. 1 and 2 have been obtained from a single synthetic DNA hairpin on multiple field-of-views. For the quantification of splicing isoforms presented in Fig. 3, the results are derived from three technical replicates from three independent biological replicates. The standard deviation represents variation among biological replicates. The mapping of epigenetic modifications on RNA presented in Fig. 4 as also been obtained from a single synthetic harpin (depicted in panel a) and the experiments were recorded over multiple fields-of-view. For Figs. 5 and 6, the number of replicates is indicated in the legend of the figure.

**Reporting summary.** Further information on research design is available in the Nature Research Reporting Summary linked to this article.

## Data availability
All the source and sequencing data can be found in the file entitled Supplementary Data 2. Further data that support the findings presented in this paper are available from the corresponding author upon reasonable request.

## Code availability
All the code used for the work presented in this paper is freely available at this address: gitlab.com/depixus-code.

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

## Acknowledgements

We would like to thank all the members of the Depixus team and François-Xavier Lyonnet du Moutier for their inputs and fruitful discussions on different aspects of the work presented here, as well as Jean Baptiste Boulé for his insight on 8-oxoG. We would like to thank Harold Gouet and Fangyuan Ding for their initial observations of epigenetic detection using antibodies on the MTs. This work was supported by a Eurostars grant entitled "RNA EPIGENETIX: Developing new tools for epigenetic analysis of RNA molecules using the SIMDEQ sequencing platform" (Reference Number: 9830), an ANR grant entitled MuSeq and a Horizon 2020 grant entitled "CONAN II-COmplete Nucleic acid ANalysis at genome scale at ultra-high throughput Phase 2" (EIC-SMEInst-2018-2020 grant # 829965). Part of this work was also supported by the European Research Council grant Magreps [267 862].

## Author contributions

T.V., G.R., V.C., D.B., and J.F.A. were involved in the development and design of the SDI platform and for its initial testing. J.O. and R.M. performed the analysis of DNA base modification, as well as participated in the determination of precision of the SDI platform. Z.W. and A.L. performed experiments on RNA base modification detection, as well as the analysis of RNA using oligonucleotides. J.M. and L.G. performed the *E. coli* and human fragment enrichment. P.D'A., D.S., and S.A.M. participated in the statistical analysis of the results, as well as data treatment. T.V., V.C., D.B., J.F.A., C.A., G.S., J.O., and G.H. participated in the design of the experiments. G.S., C.A., G.H., J.O., J.M., and Z.W. wrote the manuscript with inputs from all the other authors.

## Competing interests

J.M., Z.W., L.G., A.L., T.V., G.R., P.D'A., D.S., S.A.M., R.M. G.S., C.A., and J.O. are employees of the company Depixus SAS, which holds exclusive licenses from France's Centre National de la Recherche Scientifique (CNRS) for the use of the magnetic tweezers for sequencing and epigenetic detection. G.H. is the President (CEO) of Depixus SAS. G.H., V.C., D.B., and J.F.A. are shareholders in Depixus SAS.
