## [Peer Review File · Communications Biology]

Reviewers' comments:

Reviewer #1 (Remarks to the Author):

Generally, I think this is a very nice work with creative ideas, solid data, and interesting findings. I recommend it to be published after major revisions. Compare to their paper published in 2012 (Ref 22), there are two remarkable new improvements. 1) To analyze the chemical modifications of native DNA at specific gene loci without amplification, the authors developed a DNA enrichment method at high specificity and efficiency. 2) Unzipping and re-zipping the native DNA inside a hairpin molecule repetitively, the authors developed an assay to characterize multiple kinds of base modifications of the same DNA or RNA molecule using antibodies at high specificity and sensitivity.

The readability of the manuscript, however, needs to be greatly improved. This is partially due to the complexity of the content and is partially due to the authors' writing. My overall advice is that, please focus more on detecting multiple kinds of base modifications of DNA and RNA, which is a difficult task but the major progress in this work, rather than detecting their sequences.

The following are my concerns and suggestions the authors may take into consideration.

1. To analyze the base modifications of the native DNA at specific gene loci without amplification, the authors described a CRISPR/Cas9 enrichment method to achieve a high degree of target specificity (enrichment greater than 140, 000-fold, as pointed out in the last paragraph on Page 17). This enrichment efficiency is much higher than those of the recently amplification-free Cas9-based enrichment strategies which only achieved an enrichment of 20- to 60-fold (Ref. 10-12). Please give the details of how the number of enrichment of "140, 000-fold" was calculated, and please explain clearly why a dramatic higher enrichment efficiency could be achieved compared with previous strategies. Please also add a flowchart to supplementary figure 4 to describe the steps on how to enrich native DNA at specific loci and how the DNA fragment is constructed into a DNA hairpin for magnetic tweezers. Due to its importance and potential impacts on the field, please discuss more the applications of this super-efficient CRISPR/Cas9 enrichment method as well.

2. Using the one-micrometer-diameter bead (Dynabeads™ MyOne™ Streptavidin T1), usually <10 pN force can be archived with a pair of NdFeB magnets. Please explain why larger magnetic forces were archived in this work and what the maximum magnetic force can be archived.

3. Table 1 shows that the specificity of DNA base detection with MTs using the antibody-based approach was very high but there's still some room to increase the sensitivity. For example, 5mC is the most common and important DNA modification in eukaryotes. Is it possible to further increase the sensitivity of 5mC detection by increasing the concentration of antibody or using antibodies from some other vendors? In mammals, most 5mC occurs at CpG sites (a cytosine nucleotide followed by a guanine nucleotide), and thus it might be useful to show the specificity and sensitivity of the detection of CpG methylation in addition to the detection of 5mC.

4. May be clearer figures will be submitted just before publish, but current figures in the main text as well as in the supplementary material have poor resolution. Since different colors were used to mark different DNA fragments, it is hard to use different colors to differentiate DNA and RNA strands. I suggest using dashed/dotted lines to represent RNA strands in the figures.

5. It is a little strange that a single four-base oligonucleotide (CAAG) can be annealed to its complementary ssDNA strand (and three-base oligonucleotide can be annealed to its complementary ssDNA) and block the re-zipping of an opened hairpin. Usually, a dsDNA longer than 8 bp can stay stable base-paired at room temperature. Also, in the previous paper from the

same group (Ref. 22), they used seven- and ten-base oligonucleotide to block the re-zipping of the opened DNA hairpin. Could the author somehow explain why such short oligonucleotide can block the re-zipping of an opened hairpin? What were the unzipping and re-zipping forces cycled and what's the blocking time and frequency with the short oligonucleotide? If the re-zipping force is not low enough, some reannealing steps may also be observed without blocking oligonucleotide at AT-rich locations. Were these reannealing steps troublesome and how to avoid these troubles? Please add representative raw data (similar to Figure 1b) to the supplementary material.

6. The title. There were no experiments for the native RNA in this work. Thus, please modify the title to avoid misunderstanding.

7. Page 5, last paragraph. "to characterize both the underlying structural variation and epigenetic modification of single molecules of native DNA molecules from the clinically important gene, FMR1." Please specify what "structural variation" means exactly and whether it is characterized in this work.

8. Page 10, Paragraph 2. "A systemic difference (of 5 bp) in the position of the antibody blockage compared to the actual position of the RNA base modification, which is attributed to differential stretching of RNA compared to DNA". Does "differential stretching" means different force-extension response of RNA compared with that of DNA? If a longer or short RNA was used should the systemic difference change or not? I suggest the authors add a sketch in the supplementary material to explain the systemic differences more clearly.

9. Page 12, Paragraph 1. Is there any database or previous papers to compare the level of methylation modifications at the measured loci? Are the single-molecule results agree with previous results?

10. Page 13, Paragraph 2. I read this paragraph and its supplementary figure again and again but still couldn't understand how the three-way junction prevented the formation of the secondary structures of the CGG repeats. Please explain it in more details and modify the supplementary figure 6.

11. Page 15, Paragraph 1. Does "the consensus level" means "averaged over a big number of molecules"? Please make sure this phrase is used properly.

12. From Page 15 to 16, the authors compare this work with other sequencing methods. To be more understandable, I think these comparisons can be listed in a table.

13. Page 20. Paragraph 2. Since the way the authors prepared the DNA hairpin for DNA epigenetic detection is a little special, please add a flowchart in the supplementary material to help the readers to understand. You may refer to the flowcharts in a previous work by Yang et al ("A Universal Assay for Making DNA, RNA, and RNA-DNA Hybrid Configurations for Single-Molecule Manipulation in Two or Three Steps without Ligation". ACS synthetic biology 8:1663-1672) and Flávia S Papini et al ("High-yield fabrication of DNA and RNA constructs for single molecule force and torque spectroscopy experiments", Nucleic Acids Research 47: e144) as examples to draw your flowcharts.

14. Page 20-21. Since the way the authors prepared the DNA hairpin and flow-cell for MT is a little special, please add a flowchart in the supplementary material to help the readers to understand.

15. Page 22, Paragraph 1. The authors described "the synthetic RNA/DNA hybrid fragment was ligated to both the 600 and 400 bp DNA fragments before ligating a loop (PS359) and Y shape (triple-biotin/PS866) at either end using T3 DNA ligase (NEB)." According to the manual from NEB, T3 DNA ligase can catalyze ligation of nicks, cohesive ends, and blunt ends in dsDNA. Are there any documents, references, or experimental data supporting that T3 DNA ligase can also catalyze

ligation RNA/DNA hybrid to dsDNA? Also, please add a flowchart in the supplementary material to help the readers to understand the hairpin construction for RNA epigenetic detection.

16. Page 25, the last paragraph. I suspect whether 2'-O-Me RNA behave the same as the normal RNA here. Can the NaOH step be skipped or replaced by heating or Urea or enzyme or force-rapture? If skip the NaOH step, it seems to me that, even though there may be some wrong tethers, the analysis of RNA can still be accomplished.

17. Page 27-28. Due to its complexity, I suggest adding a step-by-step flowchart in the supplementary material for the Enrichment of E. coli targets.

18. Page 29-30. Due to its complexity, I suggest adding a step-by-step flowchart in the supplementary material for the Enrichment of FMR1 on human genomic DNA.

19. Figures 2b, 3b, and 4a. Could you give some explanation for the extra unspecific peaks in the Rate-position curves? Can you estimate the sequences of the extra peaks due to the <1 bp high precision of your method?

20. Figure 5a. What do the red dot and the yellow star represent in the hairpin assembly step?

21. Figure Supplemental figure 1. A way to show the precision of the z-axis of the microbead and the stability of the temperature is the raw long time-trace of multiple reference beads stuck to the glass surface of the flow-cell in the same camera view. Thus, could you please add this raw data?

22. Figure Supplemental figure 2. The 2nd round of PCR, the right Column, should the upper stranded in the DNA be in red rather than black? I suggest also include the last steps to finally assembly the sticky ends of the hairpin in the flowchart.

23. Figure Supplemental figure 6. Many more details are needed to explain the design and workflow of this experiment. For example, the cyan color is used to represent too many objects in the figure. The repeated position is not shown in panel (A).

24. A careful proofread is needed to polish the English. The following are some texts that need to be modified. As I am not a native English speaker, the author can modify them in a better way than my suggestions.

- 1) Introduction, Paragraph 1. In my opinion, the conversion of RNA to cDNA "lost the information of" the numerous modifications on RNA bases, rather than "erases".
- 2) Page 9, Paragraph 2. "Fig 3c" should be "Figure 3c".
- 3) Use "double-stranded" to replace "double stranded" and use "single-stranded" to replace "single stranded".
- 4) Page 14, last paragraph. "figure 6b" should be "figure 6b".
- 5) Page 16, Paragraph 1. "25x" and "250x" should be "25x" and "250x".
- 6) Page 20, Paragraph 2. "150mM NaCl" should be "150 mM NaCl".
- 7) Page 21, Paragraph 2. "500nM" should be "500 nM".
- 8) Page 25, last paragraph. "20mM" and "50mM" should be "20 mM" and "50 mM".
- 9) Page 27, Paragraph 1. "2" in "MgCl2" should be subscripted.
- 10) Page 27, Paragraph 2. "0.8 x", "200µM", "1 x", "1mM" and "20nM" should be corrected.
- 11) Page 29. "0.8 x" and "1 x" should be corrected.

Reviewer #2 (Remarks to the Author):

This work describes a platform and workflow for genomic and epigenomic analysis of targeted samples based on magnetic tweezers. The sample nucleic acids were either synthetically prepared or extracted from cells, converted into a hairpin construct, and subjected to force protocols in the

presence of molecules that interfere with hairpin re-zipping. The results are carefully analyzed and overall very impressive. There is certainly a great advancement from the previous report (Ding et al., Nat. Meth. 2012) particularly with regard to its versatility and precision. Like other sequencing techniques, the developed method is unlikely to be the one-size-fits-all technology, but I agree with the authors claim that it certainly fills the gaps like long-range profiling and multiplexed measurements on single molecules. I strongly recommend its publication in Communications Biology, mostly in its current form, with the minor points below addressed:

p9) "We observed that modifications to the oligonucleotide backbone structure and the incorporation of intercalators significantly improved binding stability." I couldn't find how this is done in the manuscript, could you please describe the details of this method?

Also, when analyzing RNA-DNA hybrid, the authors employed a synthetic "2'-O-Me RNA" sample to avoid hydrolysis (p25). If this is an essential feature, it will preclude the direct analysis of RNA sample extracted from cells. I think some clarification is needed here.

October 10th, 2020

Subject: Resubmission of the manuscript COMMSBIO-20-1573-T – response to reviewers' comments

Dear Communications Biology team and reviewers,

We thank both reviewers for their careful reviews of our manuscript and the detailed comments that they have made.

In the following document we respond to these comments and highlight the corresponding changes we have made to the manuscript.

Yours faithfully,

Gordon Hamilton

Reviewer #1:

Generally, I think this is a very nice work with creative ideas, solid data, and interesting findings. I recommend it to be published after major revisions. Compare to their paper published in 2012 (Ref 22), there are two remarkable new improvements. 1) To analyze the chemical modifications of native DNA at specific gene loci without amplification, the authors developed a DNA enrichment method at high specificity and efficiency. 2) Unzipping and re-zipping the native DNA inside a hairpin molecule repetitively, the authors developed an assay to characterize multiple kinds of base modifications of the same DNA or RNA molecule using antibodies at high specificity and sensitivity.

The readability of the manuscript, however, needs to be greatly improved. This is partially due to the complexity of the content and is partially due to the authors' writing. My overall advice is that, please focus more on detecting multiple kinds of base modifications of DNA and RNA, which is a difficult task but the major progress in this work, rather than detecting their sequences.

- **OUR RESPONSE:** we hope that our latest round of edits has succeeded in improving the readability of the manuscript. We have also, as suggested, attempted to focus the latest version of the manuscript a little more on base modification detection and a little less on sequence detection.

The following are my concerns and suggestions the authors may take into consideration.

1. To analyze the base modifications of the native DNA at specific gene loci without amplification, the authors described a CRISPR/Cas9 enrichment method to achieve a high degree of target specificity (enrichment greater than 140, 000-fold, as pointed out in the last paragraph on Page 17). This

enrichment efficiency is much higher than those of the recently amplification-free Cas9-based enrichment strategies which only achieved an enrichment of 20- to 60-fold (Ref. 10-12). Please give the details of how the number of enrichment of “140, 000-fold” was calculated, and please explain clearly why a dramatic higher enrichment efficiency could be achieved compared with previous strategies.

- **OUR RESPONSE:** We thank the reviewer for pointing this out. This number was calculated using the qPCR data from *E. coli* data by dividing the percentage of remaining molecules after the dCas9 step (average 50%) by the “outside” targets (non-protected targets, 0.13%). This gives an enrichment of 390-fold. Then, this number was multiplied by the total number of beads obtained (for the coverage, 359 beads). However, after reviewing the literature on how to calculate enrichment factors (the reference paper cited by all the studies is F. Mertes et al., “*Targeted enrichment of genomic DNA regions for next-generation sequencing*,” *Brief. Funct. Genomics*, vol. 10, no. 6, pp. 374–386, 2011.), we realize that calculating the fold-enrichment by this method is not possible as we obtained 100% specificity, which would in turn give us an infinite enrichment for both our *E. coli* and human enrichment experiments (we never observed any other sequences than the expected protected sequence). We have therefore changed the text accordingly and here is the new sentence included in the manuscript at page 18:

“To analyze base modifications and genomic variation of specific genomic loci, we developed a CRISPR/Cas9 enrichment method that can isolate native, high molecular-weight fragments of DNA from complex genomes with exceptionally high target specificity (in this paper we demonstrate 100% on-target rates). This high specificity comes from the combination of three highly specific steps in our protocol: the Cas12a step, dCas9 step, and finally the use of specific oligonucleotides to assemble the hairpins. Even if there are off-targets for these Cas proteins, a region must be flanked by two closely spaced off-targets to be protected from degradation, an unlikely occurrence especially when few targets are enriched.” (lines 386 to 393).

2. Please also add a flowchart to supplementary figure 4 to describe the steps on how to enrich native DNA at specific loci and how the DNA fragment is constructed into a DNA hairpin for magnetic tweezers.

- **OUR RESPONSE:** this has been created, added, and can be found on page 5 of the Supplementary Information (new Supplementary Figure 5).

3. Due to its importance and potential impacts on the field, please discuss more the applications of this super-efficient CRISPR/Cas9 enrichment method as well.

- **OUR RESPONSE:** following directly on from the paragraph quoted above, we have added the following text to respond to this important comment:

“The method has the benefit of not being affected by genetic variability within target regions. Furthermore, it is not only compatible with our hairpin-based MT assays but, by ligating alternative adaptors, can also be used with both NGS and other single-molecule sequencing platforms. We envisage it will be applied to the analysis of specific disease-related genetic regions where sequence and epigenetic variability play important pathogenic roles. Also, the high specificity of the approach may help to reduce the need for costly deep sequencing and see it brought into diagnostic pipelines.” (lines 394 to 400).

4. **Using the one-micrometer-diameter bead (Dynabeads™ MyOne™ Streptavidin T1), usually <10 pN force can be archived with a pair of NdFeB magnets. Please explain why larger magnetic forces were archived in this work and what the maximum magnetic force can be archived.**

- **OUR RESPONSE:** The force generated on the MyOne™ beads depends on three main features:
 - The type of magnets used: we use a pair of NdFeB magnets
 - The gap between the two magnets: we set this approximately 250 μm.
 - The distance between the beads and the magnet: in our current flow-cell setup we can approach the magnet to a distance of 110 μm from the beads.

With these three features, it is possible to generate up to 25 pN of force on MyOne™ beads. This is well documented in the literature and here are a few references (all included in the manuscript):

- Ding, F. et al. Single-molecule mechanical identification and sequencing. *Nat. Methods* 9, 367–72 (2012).
- Manosas, M. et al. Mechanism of strand displacement synthesis by DNA replicative polymerases. *Nucleic Acids Res.* 40, 6174–6186 (2012)
- Manosas, M., Spiering, M. M., Ding, F., Croquette, V. & Benkovic, S. J. Collaborative coupling between polymerase and helicase for leading-strand synthesis. *Nucleic Acids Res.* 40, 6187–6198 (2012)

5. **Table 1 shows that the specificity of DNA base detection with MTs using the antibody-based approach was very high but there's still some room to increase the sensitivity. For example, 5mC is the most common and import DNA modification in eukaryotes. Is it possible to further increase the sensitivity of 5mC detection by increasing the concentration of antibody or using antibodies from some other vendors?**

- **OUR RESPONSE:** We have tested various antibody clones against 5mC and, at the time, the Diagenode clone ICC/IF was the one offering the best balance between sensitivity and specificity (although there was some cross-reactivity). The sensitivity of the 5mC antibody clone is more than 98%, when applying thresholding (Table 1). It may be possible to further improve on this as we continue to explore the properties of other 5mC antibodies in our system. As we report for 6mA in the manuscript, using more than one antibody can improve the sensitivity of base modification detection. Unfortunately, the sensitivity of detection cannot be increased by simply continuing to increase antibody concentration as we find that at higher concentrations antibodies tend to bind non-specifically to non-modified DNA bases and degrade data quality.

6. **In mammals, most 5mC occurs at CpG sites (a cytosine nucleotide followed by a guanine nucleotide), and thus it might be useful to show the specificity and sensitivity of the detection of CpG methylation in addition to the detection of 5mC.**

- **OUR RESPONSE:** when we designed the sequence of the synthetic fragment used for antibody testing, we incorporated the 5mC modification within a CpG sequence (sequence : 5' P- tgcata1ggctgtgc 3', where 1 represents the 5mC modified base) to indeed mimic the most prevalent situation in mammalian DNA. So, the presented specificity and sensitivity in Table 1 is in the context of a CpG. However, it is worth pointing out that the sensitivity of 5mC detection does not change when the sequence context is CC*wGG (the sequence motif methylated in *E. coli* by the Dcm methylase). We see 100% sensitivity of detection on a hairpin constructed from a plasmid with two sites of this sequence (data not shown in the manuscript).

7. **May be clearer figures will be submitted just before publish, but current figures in the main text as well as in the supplementary material have poor resolution. Since different colors were used to**

mark different DNA fragments, it is hard to use different colors to differentiate DNA and RNA strands. I suggest using dashed/dotted lines to represent RNA strands in the figures.

- **OUR RESPONSE:** Many thanks for pointing this out. We have generated new figures with higher resolution for this resubmission and, as suggested, have changed the way we represent RNA vs DNA strands on the figures.
8. It is a little strange that a single four-base oligonucleotide (CAAG) can be annealed to its complementary ssDNA strand (and three-base oligonucleotide can be annealed to its complementary ssDNA) and block the re-zipping of an opened hairpin. Usually, a dsDNA longer than 8 bp can stay stable base-paired at room temperature. Also, in the previous paper from the same group (Ref. 22), they used seven- and ten-base oligonucleotide to block the re-zipping of the opened DNA hairpin. Could the author somehow explain why such short oligonucleotide can block the re-zipping of an opened hairpin?

- **OUR RESPONSE:** Indeed, at room temperature, plain DNA oligonucleotides shorter than eight bases do not hybridize with hairpins strongly enough to cause detectable blockages of the fork during refolding of the hairpin. We have now provided in the text more detail on how the oligonucleotides have been chemically modified to make their hybridisation compatible fork blockage at room temperature. We have written:

“We observed that modifying the oligonucleotide backbone structure with locked nucleic acid (LNA) and the incorporation of intercalators, such as acridine orange, significantly improved their binding stability and allowed us to detect their binding with our platform.” (lines 195 to 198)

9. What were the unzipping and re-zipping forces cycled and what’s the blocking time and frequency with the short oligonucleotide?

- **OUR RESPONSE:** The force used to open and close hairpins is independent of the type of oligonucleotides used (i.e. whether 3 bases or 10 bases in length). Specifically, we aim to open hairpins by applying 20-25pN of force and let them re-zip by reducing the force to between 10-13pN.

To answer the second part of your question, we have generated a Supplemental Figure 9 showing the average binding time and hybridization rate for all the positions on all the four different hairpins enriched from *E. coli* gDNA for which the oligonucleotide CAAG was used for identification.

10. If the re-zipping force is not low enough, some reannealing steps may also be observed without blocking oligonucleotide at AT-rich locations. Were these reannealing steps troublesome and how to avoid these troubles?

- **OUR RESPONSE:** What you describe could be an issue if the closing force is very close to the hysteresis of the hairpin. However, the binding kinetics of these short oligonucleotides allowed us to use a closing force away from this position and therefore any stretch of weak bases that could potentially cause a blockage without any oligonucleotide can be avoided. In short, these reannealing steps were not troublesome to us.

11. Please add representative raw data (similar to Figure 1b) to the supplementary material.

- **OUR RESPONSE:** Supplemental Figure 10 now presents examples of traces obtained from the use of the oligonucleotide CAAG on the four human targets for their identification.

12. The title. There were no experiments for the native RNA in this work. Thus, please modify the title to avoid misunderstanding.

- **OUR RESPONSE:** This is a good point. We agree that the title was misleading due to the “and”. Given this, and the 15-word limit, we have changed the title to “*Detection of genetic variation and base modifications at base-pair resolution on both DNA and RNA*”.

13. Page 5, last paragraph. “to characterize both the underlying structural variation and epigenetic modification of single molecules of native DNA molecules from the clinically important gene, FMR1.” Please specify what “structural variation” means exactly and whether it is characterized in this work.

- **OUR RESPONSE:** The term ‘structural variation’ is used here to describe the presence of the trinucleotide repeats in the 5’ UTR of the Fmr1 locus. We have shown in this manuscript that we can use magnetic tweezers to characterize the lengths of these repeats. Not that short tandem repeats (STRs), which include trinucleotide repeat like those found in the Fmr1 gene, can be referred to as structural variation according to NCBI website:

https://www.ncbi.nlm.nih.gov/dbvar/content/var_summary/

For clarity, we the relevant sentence in the manuscript now reads:

“to characterize both the underlying structural variation (trinucleotide repeats) and epigenetic modification of single molecules of native DNA molecules from the clinically important gene, FMR1” (lines 114 to 116).

14. Page 10, Paragraph 2. “A systemic difference (of 5 bp) in the position of the antibody blockage compared to the actual position of the RNA base modification, which is attributed to differential stretching of RNA compared to DNA”. Does “differential stretching” means different force-extension response of RNA compared with that of DNA? If a longer or short RNA was used should the systemic difference change or not? I suggest the authors add a sketch in the supplementary martial to explain the systemic differences more clearly.

- **OUR RESPONSE:** These are excellent questions. Indeed, our mention of ‘differential stretching’ refers to the different force-extension responses between a single RNA and DNA base. It has been shown that ssDNA and ssRNA have different stiffnesses (Jacobson et al. 2017. “Single-stranded nucleic acid elasticity arises from internal electrostatic tension”. PNAS, 114 (20) 5095-5100), and therefore at a given force, the z-extension is shorter for ssRNA compared with ssDNA (cf. formula [1] in paper). This is an intrinsic property of DNA or RNA, and therefore the systematic different would not change, whether a longer or shorter RNA molecule is used. Based on the data we present, the difference between DNA and RNA stretch seems to strongly contribute to the 5 bp difference we observed in the z-position compared with theoretical position of the modification on the molecule.

To quantify this and correct for this difference, we used an internal control (oligonucleotides, which terminate at the position of the modification). Using this strategy, we were able to reduce the detected difference between the mapping of the oligonucleotide and the antibody to less than 2 bases.

To clarify this, we have changed the sentence to read: “*suggesting that this was due to the differential force-extension response between single bases of RNA compared to DNA²⁵⁻²⁷”* (lines 235 & 236)

15. Page 12, Paragraph 1. Is there any database or previous papers to compare the level of methylation modifications at the measured loci? Are the single-molecule results agree with previous results?

- **OUR RESPONSE:** There is no database or previous paper that describes methylation at the single molecule level for the DNA sample (NA06896). Typically, methylation analysis of Fmr1 is performed through subtractive PCR, where the genomic DNA is first cut with a methylation sensitive restriction enzyme (HpaII) and PCR is done on the uncut versus cut sample (Chen et al., “High-resolution methylation polymerase chain reaction for fragile X analysis: Evidence for novel FMR1 methylation patterns undetected in Southern blot analyses,” *Genet. Med.*, vol. 13, no. 6, pp. 528–538, 2011). This methylation site corresponds to the CpG site n°65 in our study (position 147911956 in the chromosome X).

We compared our data with the analysis of this clinical sample in the mentioned paper. If we only consider this position, we observed that 36% of normal alleles showed methylation at this CpG compared to 21% in the published literature. Moreover, we observed that 83% of the molecules we identified as having between 110-149 repeats are methylated compared to 82.5% in this reference. Finally, we did not observe methylation on molecules having more than 150 repeats for this position, which agrees with what is reported in the literature.

Due to word limit constraints, we have not been able to go into a lot of detail on this point in the text, but we have written:

“Our results were in concordance with Chen et al. (2011) who used a methylation-specific PCR test and reported a lack of methylation on repeats greater than 150, but high levels of methylation for pre-mutation alleles and low levels for normal repeats³⁴.” (lines 329 to 331)

16. Page 13, Paragraph 2. I read this paragraph and its supplementary figure again and again but still couldn't understand how the three-way junction prevented the formation of the secondary structures of the CGG repeats. Please explain it in more details and modify the supplementary figure 6.

- **OUR RESPONSE:** Many thanks for pointing out the difficulty with understanding this assay. We have attempted to make everything clearer with an updated Supplemental Figure 11.

In summary, during our standard experiment where we monitor blockages during the closing phase, oligonucleotides bind and block the refolding of the fork at low force (ca. 10-12 pN). However, during this closing phase at low force, secondary structures can form in the repeated region (including intra-strand hairpin formation: Marquis Gacy, A., Goellner, G., Jurani, N., Macura, S., & McMurray, C. T. (1995). *Trinucleotide repeats that expand in human disease form hairpin structures in vitro*. *Cell*, 81(4), 533–540. [https://doi.org/10.1016/0092-8674\(95\)90074-8](https://doi.org/10.1016/0092-8674(95)90074-8)). This shortens the molecule, generates biases in the positions of detected blockages.

In order to overcome these secondary structures, we developed a strategy where we block the fork during the high force phase (>20pN) such that any secondary structures present will be unfolded. We used specific oligonucleotides that contains a single-stranded region followed by a loop structure composed of the sequence following the region where the ssDNA region hybridizes. The single stranded region complementary to the hairpin will bind during the opening of the hairpin and block the fork during the re-zipping of the hairpin (as our normal assay). Because the stem loop on the oligonucleotide contains the same sequence as the hairpin, strand invasion will occur and allow the formation of a three-way junction. When the force is increased above 20pN, the opening of the hairpin is transiently prevented by this three-way junction. During these blockages at high force, any

secondary structure present within the CGG repeats will unfold, and the full extension of the single stranded molecule is measured with high precision. Eventually, the flapping part of the oligonucleotide will be ejected by the high shearing force and the hairpin will open completely.

This assay allows us to block the opening of the hairpin at multiple positions within the hairpin by using several oligonucleotides on each side of the repeat position (similar to the closing assay). These oligonucleotides are extremely specific for the sequence of the hairpin, as for the closing assay. However, due to the nature of the blocking oligonucleotide, we can only get one blocking position per cycle as opposed to multiple positions with our closing assay. As for the closing assay, we can align the blocking positions outside the repeat region (which is invariant) to calculate the stretch of the molecule and apply this stretch to the measured distance between the two blocking positions directly flanking the repeat to calculate the number of bases present. The three-way junction oligonucleotides are therefore not binding on the repeat *per se* but allow us to stretch the molecules at high force to eliminate any secondary structure for accurate measurements.

17. Page 15, Paragraph 1. Does “the consensus level” means “averaged over a big number of molecules”? Please make sure this phrase is used properly.

- **OUR RESPONSE:** We thank the reviewer for pointing this out. We have tried to tighten up the text here and now talk about a ‘consensus sequence accuracy’ which is generated by aligning all the molecules that were sequenced and then determining the percentage of bases that were called correctly in at least in 50% of the molecules.

We have updated the text according to your comment and here is the revised sentence:

“The consensus sequence accuracy for these molecules was 95%, under the criterion that the correct base was called in the correct position in >50% of the molecules (Figure 3c).” (lines 207 to 209).

18. From Page 15 to 16, the authors compare this work with other sequencing methods. To be more understandable, I think these comparisons can be listed in a table.

- **OUR RESPONSE:** We attempted to put this suggestion into practice but did not find a simple formulation for such table that avoids reasonable amounts of explanatory text. We have therefore left the text, for now, unchanged, but we would be happy to revisit this again if the reviewer feels strongly about this.

19. Page 20. Paragraph 2. Since the way the authors prepared the DNA hairpin for DNA epigenetic detection is a little special, please add a flowchart in the supplementary material to help the readers to understand. You may refer to the flowcharts in a previous work by Yang et al (“A Universal Assay for Making DNA, RNA, and RNA–DNA Hybrid Configurations for Single-Molecule Manipulation in Two or Three Steps without Ligation”. ACS synthetic biology 8:1663-1672) and Flávia S Papini et al (“High-yield fabrication of DNA and RNA constructs for single molecule force and torque spectroscopy experiments”, Nucleic Acids Research 47: e144) as examples to draw your flowcharts. From High-yield fabrication of DNA and RNA constructs for single molecule force and torque spectroscopy experiments

- **OUR RESPONSE:** As requested, a supplemental figure has been produced to guide the reader on the procedure to make the synthetic hairpins for epigenetic analysis (Supplemental Figure 3).

20. Page 20-21. Since the way the authors prepared the DNA hairpin and flow-cell for MT is a little special, please add a flowchart in the supplementary material to help the readers to understand.

- **OUR RESPONSE:** As requested, a supplemental figure has been produced to guide the reader on the procedure to make these hairpins and attach them to the surface (Supplemental Figures 1 & 2).

21. Page 22, Paragraph 1. The authors described “the synthetic RNA/DNA hybrid fragment was ligated to both the 600 and 400 bp DNA fragments before ligating a loop (PS359) and Y shape (triple-biotin/PS866) at either end using T3 DNA ligase (NEB).” According to the manual from NEB, T3 DNA ligase can catalyze ligation of nicks, cohesive ends, and blunt ends in dsDNA. Are there any documents, references, or experimental data supporting that T3 DNA ligase can also catalyze ligation RNA/DNA hybrid to dsDNA?

- **OUR RESPONSE:** We appreciate the reviewer having taken the time to think through the detailed enzymatic step we have used for ligation during hairpin preparation. Indeed, T3 DNA ligase is capable of ligating DNA ends, but not RNA ends. In our synthetic hairpin preparation, however, we ordered the flanking RNA fragments with DNA bases at their extremities in contact with the dsDNA fragment (Supplemental Table 2). Therefore, after annealing, we only have DNA sticky ends to be ligated to DNA fragments.

We use T3 DNA ligase instead of T4 DNA ligase because we have found out that T4 DNA ligase binds to RNA fragments non-specifically and therefore titrates out the effective enzyme concentration. To make the text clearer, we have changed the text into: *“the resulting synthetic RNA/DNA hybrid fragments with DNA overhang was ligated...”* (lines 492 & 493). The overall hairpin construction is the same as Supplementary Figure 1, and the different components are indicated in the legend of Figure 4a.

22. Also, please add a flowchart in the supplementary material to help the readers to understand the hairpin construction for RNA epigenetic detection.

Since the construction of both RNA and DNA synthetic hairpins follows the same protocol, we can avoid the need for the additional flowchart requested here. We clarify this in Supplementary Figure 1 with the following figure legend: *“Strategy for construction of synthetic RNA and DNA hairpins containing epigenetic modifications from chemically synthesized oligonucleotides”*

23. Page 25, the last paragraph. I suspect whether 2'-O-Me RNA behave the same as the normal RNA here. Can the NaOH step be skipped or replaced by heating or Urea or enzyme or force-rapture? If skip the NaOH step, it seems to me that, even though there may be some wrong tethers, the analysis of RNA can still be accomplished.

- **OUR RESPONSE:** We thank the reviewer for raising the concern for NaOH treatment and the usage of 2'-O-Me RNA. We have tested the behaviour of several short oligonucleotides in short stretches of RNA bases compared with 2'-O-Me RNA bases and did not find any significant differences in hybridization rate and time (data not shown in this manuscript). We have decided to use 2'-O-Me RNA because these oligos are easier to handle and more stable for long-term experiments. The NaOH step can indeed be skipped, and in fact when working with native RNA molecules we avoid any NaOH treatment. To clarify this, we have changed the text to *“The methylation group of 2'-O-Me RNA bases protects the ribose from hydrolysis, allowing the washing step with NaOH. However, this washing step is not performed when hairpins containing natural RNA bases are used.”* (lines 583 to 585).

24. Page 27-28. Due to its complexity, I suggest adding a step-by-step flowchart in the supplementary material for the Enrichment of E. coli targets.

- **OUR RESPONSE:** To expand on the simplified version of the step-by-step workflow in Figure 5a, we have put together a more detailed version in Supplemental Figure 8.

25. Page 29-30. Due to its complexity, I suggest adding a step-by-step flowchart in the supplementary material for the Enrichment of FMR1 on human genomic DNA.

- **OUR RESPONSE:** The protocol is essentially the same whether the DNA comes from *E. coli* or human. The only differences are the RNA guides for both Cas12a and dCas9 and the oligonucleotides used to assemble the hairpin, which are specific for each target. The list of RNA guides and oligonucleotides used to construct these *E. coli* and human hairpins from our enrichment protocol is provided in the supplemental material tables.

26. Figures 2b, 3b, and 4a. Could you give some explanation for the extra unspecific peaks in the Rate-position curves? Can you estimate the sequences of the extra peaks due to the <1 bp high precision of your method?

- **OUR RESPONSE:** As pointed out, these extra peaks are non-specific peaks that occur in around 1% of the total number of open-close cycles. They are stochastic in nature and come from, as yet unidentified sources. For all the figures mentioned by the reviewer, the graphs are from the same beads, yet between different experiments, most of these extra peaks do not overlap, showing that these peaks are not due to specific sequences at specific positions.

Since we set a threshold of requiring at least 3% of the cycles to have a blockage at a specific position to call a peak, these bindings are filtered out from our analysis.

We could estimate the sequence of these extra peaks based on the precision of our detection although, based on what we know of the stochastic nature of these peaks, we're unclear whether this extra information would strengthen the manuscript.

27. Figure 5a. What do the red dot and the yellow star represent in the hairpin assembly step?

- **OUR RESPONSE:** The red dot represents the Biotin at the 5' end to attach hairpin to the MyOne™ streptavidin bead. The yellow star represents a phosphate group in 3' of the oligo to prevent the polymerization with the Bst full-length DNA polymerase used during the hairpin assembly step. This information has been added to the legend to improve clarity.

28. Figure Supplemental figure 1. A way to show the precision of the z-axis of the microbead and the stability of the temperature is the raw long time-trace of multiple reference beads stuck to the glass surface of the flow-cell in the same camera view. Thus, could you please add this raw data?

- **OUR RESPONSE:** We have provided a composite figure in response to this suggestion (Supplemental Figure 4) which shows (i) temperature stability of the instrument over a long period of time, (ii) the capacity to run the system at sub-ambient temperatures down to 8 degrees, (iii) the importance of magnet thermalization to prevent thermal drift, and (iv) the trace of a fixed bead to demonstrate the gain in stability of the signal in our new instrument system versus the previous version.

29. Figure Supplemental figure 2. The 2nd round of PCR, the right Colum, should the upper stranded in the DNA be in red rather than black? I suggest also include the last steps to finally assembly the sticky ends of the hairpin in the flowchart.

- **OUR RESPONSE:** We thank the reviewer for pointing this out. The upper strand in the DNA should indeed be red as reviewer suggested. We have changed the colour accordingly and corrected the final assembly step.

30. Figure Supplemental figure 6. Many more details are needed to explain the design and workflow of this experiment. For example, the cyan colour is used to represent too many objects in the figure. The repeated position is not shown in panel (A).

- **OUR RESPONSE:** We fully agree with this point about improving colours. We have made changes to the colour scheme and legend which we hope will now give the reader a better understanding of how this repeat measurement assay works.

The repeat position within the schematic representation in Panel A was not needed to understand the principle behind the three-way junction opening assay, as this assay can be performed with any molecule as soon as the sequence is known. This is why it is not represented in panel A. However, as discussed previously in our response to comment 16, this assay is really useful when secondary structures are present to overcome them and generate accurate measurements.

31. A careful proofread is needed to polish the English. The following are some texts that need to be modified. As I am not a native English speaker, the author can modify them in a better way than my suggestions.

- 1) Introduction, Paragraph 1. In my opinion, the conversion of RNA to cDNA “lost the information of” the numerous modifications on RNA bases, rather than “erases”.
- 2) Page 9, Paragraph 2. “Fig 3c” should be “Figure 3c”.
- 3) Use “double-stranded” to replace “double stranded” and use “single-stranded” to replace “single stranded”.
- 4) Page 14, last paragraph. “figure 6b” should be “figure 6b”.
- 5) Page 16, Paragraph 1. “25x” and “250x” should be “25 \times ” and “250 \times ”.
- 6) Page 20, Paragraph 2. “150mM NaCl” should be “150 mM NaCl”.
- 7) Page 21, Paragraph 2. “500nM” should be “500 nM”.
- 8) Page 25, last paragraph. “20mM” and “50mM” should be “20 mM” and “50 mM”.
- 9) Page 27, Paragraph 1. “2” in “MgCl2” should be subscripted.
- 10) Page 27, Paragraph 2. “0.8 x”, “200 μ M”, “1 x”, “1mM” and “20nM” should be corrected.
- 11) Page 29. “0.8 x” and “1 x” should be corrected.

- **OUR RESPONSE:** our thanks for these edits. All these points within the text have been addressed and can be reviewed via the tracked changes function within Word.

Reviewer #2:

This work describes a platform and workflow for genomic and epigenomic analysis of targeted samples based on magnetic tweezers. The sample nucleic acids were either synthetically prepared or extracted from cells, converted into a hairpin construct, and subjected to force protocols in the presence of molecules that interfere with hairpin re-zipping. The results are carefully analyzed and overall very impressive. There is certainly a great advancement from the previous report (Ding et al., Nat. Meth. 2012) particularly with regard to its versatility and precision. Like other sequencing techniques, the developed method is unlikely to be the one-size-fits-all technology, but I agree with the authors claim that it certainly fills the gaps like long-range profiling and multiplexed measurements on single

molecules. I strongly recommend its publication in *Communications Biology*, mostly in its current form, with the minor points below addressed:

32. p9) "We observed that modifications to the oligonucleotide backbone structure and the incorporation of intercalators significantly improved binding stability." I couldn't find how this is done in the manuscript, could you please describe the details of this method?

- **OUR RESPONSE:** These short oligonucleotides contain commercially available modifications on both the sugar backbone (LNA) and an intercalator (acridine orange) at both extremities such that the hybridization of such short oligonucleotides is possible. These oligonucleotides were not produced by a commercial oligonucleotide synthesis company, hence why we did not include any specific information on their synthesis in this manuscript. Note that we have included the following sentence in the manuscript to provide the reader with more clarity on this: "*We observed that modifying the oligonucleotide backbone structure with locked nucleic acid (LNA) and the incorporation of intercalators, such as acridine orange, significantly improved their binding stability and allowed us to detect their binding with our platform.*" (line 195)

33. Also, when analyzing RNA-DNA hybrid, the authors employed a synthetic "2'-O-Me RNA" sample to avoid hydrolysis (p25). If this is an essential feature, it will preclude the direct analysis of RNA sample extracted from cells. I think some clarification is needed here.

- **OUR RESPONSE:** We thank the reviewer for pointing this out. We used a 2'-O-Me RNA fragment for these experiments largely because they are more stable than natural RNA bases as these experiments were long (lasting > 3 days) as all the injections are performed manually. However, this NaOH washing step is not necessary (and even avoided) when we work with native RNA fragments. Following the experiments described in this paper, we have developed buffer conditions where a ligation step and subsequent NaOH clean-up is no longer required, allowing us to work with native RNA for > 10 hours. While we are not yet ready to publish on these new protocols, we have provided a small clarification on this point in the text: "*The methylation group of 2'-O-Me RNA bases protects the ribose from hydrolysis, allowing the washing step with NaOH. However, this washing step is not performed when hairpins containing natural RNA bases are used*" (lines 583 to 585).

REVIEWERS' COMMENTS:

Reviewer #1 (Remarks to the Author):

The revised manuscript has been improved greatly. I suggest it be accepted for publication.

Reviewer #2 (Remarks to the Author):

Both of my previous points are sufficiently addressed, and the authors' response to the other referee's admirably thorough review and proofreading were faithfully addressed. I recommend is publication in Communications Biology in its current form.